# Discovering Preference Optimization Algorithms with and for Large Language Models

**Chris Lu**[*]
Sakana AI and FLAIR
chrislu@sakana.ai

**Samuel Holt**[*]
University of Cambridge
sih31@cam.ac.uk

**Claudio Fanconi**[*]
University of Cambridge
caf83@cam.ac.uk

**Alex J. Chan**[†]
University of Cambridge
ajc340@cam.ac.uk

**Jakob Foerster**[‡]
FLAIR, University of Oxford
jakob.foerster@eng.ox.ac.uk

**Mihaela van der Schaar**[‡]
University of Cambridge
mv472@cam.ac.uk

**Robert Tjarko Lange**[‡]
Sakana AI
robert@sakana.ai

## Abstract

Offline preference optimization is a key method for enhancing and controlling the quality of Large Language Model (LLM) outputs. Typically, preference optimization is approached as an offline supervised learning task using manually crafted convex loss functions. While these methods are based on theoretical insights, they are inherently constrained by human creativity, so the large search space of possible loss functions remains under-explored. We address this by performing LLM-driven *objective discovery* to automatically discover new state-of-the-art preference optimization algorithms without (expert) human intervention. Specifically, we iteratively prompt an LLM to propose and implement new preference optimization loss functions based on previously evaluated performance metrics. This process leads to the discovery of previously unknown and performant preference optimization algorithms. The best performing of these we call *Discovered Preference Optimization* (DiscoPOP)[1], a novel algorithm that adaptively blends logistic and exponential losses. Experiments demonstrate the state-of-the-art performance of DiscoPOP and its successful transfer to held-out tasks.

## 1 Introduction

Training Large Language Models (LLMs) usually involves starting with a model pre-trained on large text corpora and then fine-tuning it to match human preferences. Pre-trained, and even instruction fine-tuned LLMs, can generate harmful, dangerous, and unethical completions [Carlini et al., 2021, Gehman et al., 2020]. To mitigate this and align an LLM with human values, we use human preference alignment through preference-ranked completion data. This approach has become an industry standard, popularized by reinforcement learning with human feedback (RLHF) [Christiano et al., 2017, RLHF], and more recently, by offline preference optimization algorithms like direct preference optimization [Rafailov et al., 2023, DPO] and sequence likelihood calibration [Zhao

---

[*]Equal Contribution.
[†]Work partially done at Spotify.
[‡]Equal Advising.
[1]Code: https://github.com/luchris429/DiscoPOP.

38th Conference on Neural Information Processing Systems (NeurIPS 2024).

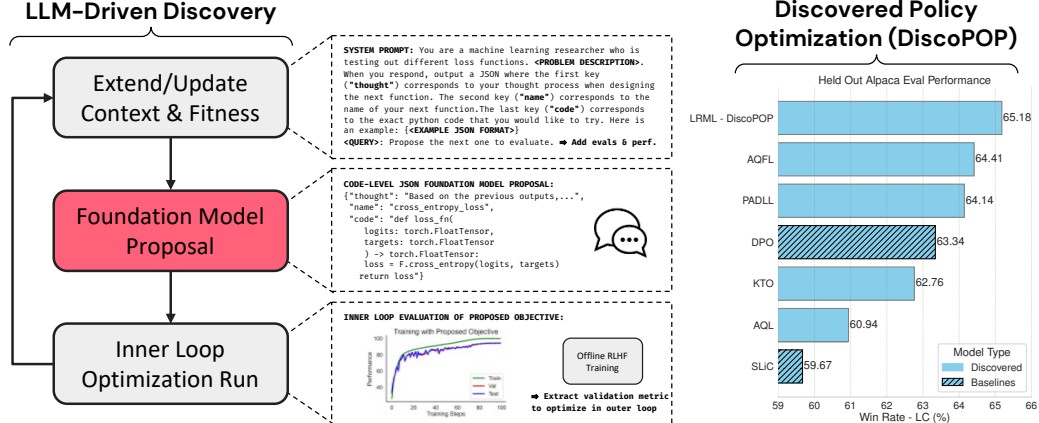

Figure 1: **Left**. Conceptual illustration of LLM-driven discovery of objective functions. We prompt an LLM to output new code-level implementations of offline preference optimization losses $\mathbb{E}_{(y_w, y_l, x) \sim \mathcal{D}} \left[ f \left( \beta \rho \right) \right]$ as a function of the policy ($\pi_\theta$) and reference model's ($\pi_{\text{ref}}$) likelihoods of the chosen ($y_w$) and rejected ($y_l$) completions. Afterwards, we run an inner loop training procedure and evaluate the resulting model on MT-Bench. The corresponding performance is fed back to the language model, and we query it for the next candidate. **Right**. Performance of discovered objective functions on Alpaca Eval.

et al., 2023, SLiC], which cast the problem as a supervised learning objective. Many algorithms have been proposed in the literature for offline preference optimization, and it remains an open question which one performs best across tasks. While a strictly dominant algorithm may not exist, some algorithms likely exhibit generally improved performance. To date, all existing state-of-the-art preference optimization algorithms [Rafailov et al., 2023, Azar et al., 2023, Zhao et al., 2023] have been developed by human experts. Despite their advancements, these solutions are inherently constrained by human limitations, including creativity, ingenuity, and expert knowledge.

In this work, we aim to address these limitations by performing LLM-driven discovery to *automatically* generate new state-of-the-art preference optimization algorithms without continual expert human intervention in the development process. While previous works [Ma et al., 2023, Yu et al., 2023] have used LLMs to design environment-specific RL reward functions, we discover general-purpose objective functions which can be used across various preference optimization tasks. More specifically, we iteratively prompt an LLM to propose new preference optimization loss functions and evaluate them, with the previously proposed loss functions and their task performance metric (in our case, MT-Bench scores [Zheng et al., 2024]) as in-context examples. After performing this automatic discovery process, we catalogue high-performing loss functions and introduce a particularly strong one we call *Discovered Preference Optimization* (DiscoPOP), a new algorithm. To ensure robustness beyond MT-Bench, we validate DiscoPOP using AlapacaEval 2.0 [Dubois et al., 2024], showing an improvement in win rates against GPT-4 from DPO ($11.23\% \rightarrow 13.21\%$). Additionally, in separate, held-out, tasks such as summarization and controlled generation, models trained with the DiscoPOP loss outperform or perform competitively with existing preference optimization algorithms.

**Contributions:** ① We propose an LLM-driven objective discovery pipeline to discover novel offline preference optimization algorithms (Section 3). ② We discover multiple high-performing preference optimization losses. One such loss, which we call *Discovered Preference Optimization* (DiscoPOP), achieves strong performance across multiple held-out evaluation tasks of multi-turn dialogue (AlpacaEval 2.0), controlled sentiment generation (IMDb) and summarization (TL;DR) tasks. ③ We provide an initial analysis of DiscoPOP, which is a weighted sum of logistic and exponential losses, and discover surprising features. For example, DiscoPOP is non-convex.

## 2 Background

**Preference Optimization**. Consider a pre-trained language model policy $\pi_\theta$ and a dataset $\mathcal{D} = \{(x^i, y_w^i, y_l^i)\}_{i=1}^N$ consisting of prompts $x$ and preference-ranked completions $y_w$ and $y_l$. In this dataset, a human rater prefers $y_w$ over $y_l$, denoted as $y_w \succ y_l$. The task is to align $\pi_\theta$ with the human values implicit in these preferences. Canonically, this has been achieved through reinforcement learning from human feedback [Christiano et al., 2017, RLHF], an approach that proceeds in two

phases: First, a *reward modelling* stage that learns a parameterized reward model $r_\phi$. By assuming a Bradley-Terry model [Bradley and Terry, 1952] of preferences, the probability of the data can be expressed as $P(y_w \succ y_l) = \exp r_\phi(y_w, x)/(\exp r_\phi(y_w, x) + \exp r_\phi(y_l, x))$, and subsequently simply optimized over $\phi$ through the maximum likelihood principle. The second stage of *policy optimization* employs a reinforcement learning algorithm to train the language model against the learned reward. Usually, a KL penalty is introduced between the model and the pre-RL *reference* policy $\pi_{ref}$ [Jaques et al., 2019, Stiennon et al., 2020] to prevent over-optimization and straying too far from the original policy, resulting in the final objective:

$$\max_{\pi_\theta} \underbrace{\mathbb{E}_{y \sim \pi_\theta, x \sim \mathcal{P}} \left[ r_\phi(y, x) \right]}_{\text{reward maximization}} - \beta \underbrace{\mathbb{KL}(\pi_\theta, \pi_{\text{ref}})}_{\text{regularization}}. \tag{1}$$

Despite success in frontier models [Anthropic, 2023, Gemini-Team, 2023], deep RL has many implementations [Engstrom et al., 2019] and training challenges [Sutton, 1984, Razin et al., 2023] that hinder its adoption. To simplify the whole process, *direct preference optimization* [Rafailov et al., 2023, DPO] aims to forego both the reward modelling and online RL procedure. Rewriting (1) with a decomposition of the KL term into:

$$\max_{\pi_\theta} \mathbb{E}_{y \sim \pi_\theta, x \sim \mathcal{P}} \left[ \underbrace{r_\phi(y, x)}_{\text{reward}} + \underbrace{\beta \log \pi_{ref}(y|x)}_{\pi_{ref} \text{ regularization}} \right] + \underbrace{\beta \mathcal{H}(\pi_\theta)}_{\text{policy entropy}}, \tag{2}$$

expresses the problem as an entropy-regularised RL bandit task [Ziebart et al., 2008], for which a known analytical solution exists: $\pi^*(y|x) = Z(x)^{-1} \pi_{ref}(y|x) \exp\left(\beta^{-1} r_\phi(y, x)\right)$. By rearranging the reward, we can express the task as a binary classification problem based on the reward difference:

$$\min_{\pi_\theta} \mathbb{E}_{(y_w, y_l, x) \sim \mathcal{D}} \left[ f \left( \beta \cdot \underbrace{\left( \log \frac{\pi_\theta(y_w|x)}{\pi_{\text{ref}}(y_w|x)} - \log \frac{\pi_\theta(y_l|x)}{\pi_{\text{ref}}(y_l|x)} \right)}_{r_\phi(y_w, x) - r_\phi(y_l, x)} \right) \right]. \tag{3}$$

Here, we define the log ratio difference as $\rho = \log \frac{\pi_\theta(y_w|x)}{\pi_{\text{ref}}(y_w|x)} - \log \frac{\pi_\theta(y_l|x)}{\pi_{\text{ref}}(y_l|x)}$. In DPO, the function $f = -\log \sigma$ is derived as the negative log of the sigmoid function given the BT model assumptions. However, Tang et al. [2024] highlighted that more generally we can obtain a recipe for offline preference optimization algorithms by letting $f : \mathbb{R} \to \mathbb{R}$ be any scalar loss function. For example, setting $f(x) = (x-1)^2$, the squared loss function [Rosasco et al., 2004] yields IPO [Azar et al., 2023], while employing the max-margin inspired hinge loss [Boser et al., 1992, Cortes and Vapnik, 1995] $f(x) = \max(0, 1-x)$ produces SLiC [Zhao et al., 2023].

**Meta-Optimization for Algorithm Discovery**. The goal of *meta-optimization* (optimizing the optimization process) is to uncover novel learning algorithms using a data-driven process. Suppose that an algorithm uses an objective function $f^\gamma$ to train a model for $K$ iterations, where $\gamma$ denotes a set of meta-parameters. Meta-optimization searches for an objective that maximizes the expected downstream performance $\max_\gamma \mathbb{E}[\eta(\pi_K)|\texttt{train}(f^\gamma)]$ where $\eta$ is a downstream performance metric. Unlike previous methods that rely on a predefined parameterization of $\gamma$ (e.g., a neural network [Hospedales et al., 2021] or domain-specific language [Alet et al., 2020]), we leverage LLMs to directly propose code-level objective functions in Python. This approach eliminates the need for a carefully designed search space and utilizes the extensive knowledge embedded in the LLM for flexible selection and mutation.

## 3   LLM-Driven Objective *Discovery*

Choosing an appropriate objective function is crucial for instilling capabilities into networks. Here, we detail our discovery process facilitated by LLM code-level objective function proposals:

**Initial Context Construction**. In the initial system prompt, we 'burn-in' the LLM using several established objective functions given in code and their corresponding performance. Furthermore, we provide problem details and an example of the output response format as a JSON dictionary.

**LLM Querying, Parsing & Output Validation**. We query the LLM, parse the response JSON, and run a set of unit tests (e.g. for valid output shapes) before starting a training run. If the parsing or unit tests fail, we resample a new solution after providing the error message as feedback to the LLM.

**Performance Evaluation**. The proposed objective function is then evaluated based on its ability to optimize a model for a predefined downstream validation task. We refer to the resulting performance metric as $\eta$.

**Iterative Refinement**. By using the performance provided as feedback, the LLM iteratively refines its proposals. In each iteration, the model synthesizes a new candidate loss function, exploring both variations of previously successful formulas and entirely new formulations that might improve upon the existing benchmarks. This iterative process is repeated for a specified number of generations or until convergence when a set of optimal loss functions is observed.

We summarise this general objective discovery process in Figure 1 and is shown in Algorithm 1.

---

**Algorithm 1** LLM-Driven Objective Discovery

1: Initialize LLM with established loss functions and their performance *in context*.
2: **repeat** for each generation $i$
3:     LLM proposes a new candidate objective function $f_i$
4:     Run unit tests to check the validity of the candidate and resample if needed.
5:     Evaluate the objective function using the performance metric $\eta$
6:     Update the LLM context with the performance data
7:     LLM refines generation strategy based on the feedback
8: **until** convergence criteria are met or maximum generations are reached

---

**Small case study: Discovering supervised classification loss functions**. Consider the case of supervised classification on the CIFAR-10 dataset as a simple starting example. We train a simple ResNet-18 for 5 epochs using the objectives proposed by GPT-4 [OpenAI, 2023]. After each training run we provide the LLM with the corresponding validation accuracy and query it for the next PyTorch-based [Paszke et al., 2017] candidate objective function.

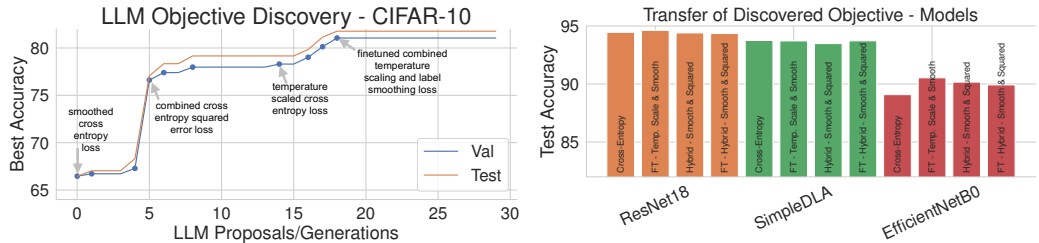

Figure 2: LLM-driven objective discovery for CIFAR-10 classification. **Left**. Performance across LLM-discovery trials. The proposals alternate between exploring new objective concepts, tuning the components, and combining previous insights. **Right**. The best three discovered objectives transfer to different network architectures and longer training runs (100 epochs).

Figure 2 depicts the performance of the proposed objective functions across the discovery process. The different discovered objectives all outperform the standard cross-entropy loss. Interestingly, we observe that the LLM-driven discovery alternates between several different exploration, fine-tuning, and knowledge composition steps: Initially, the LLM proposes a label-smoothed cross-entropy objective. After tuning the smoothing temperature, it explores a squared error loss variant, which improved the observed validation performance. Next, the two conceptually different objectives are combined, leading to another significant performance improvement. Hence, the LLM discovery process does not perform a random search over objectives previously outlined in the literature but instead composes various concepts in a complementary fashion. Furthermore, the discovered objectives also generalize to different architectures and longer training runs. In Appendix D.3 we show that this process of discovery is robust to the choice of sampling temperature and prompt/context construction.

## 4 Discovering Offline Preference Optimization Objectives

In this section, we run our LLM-driven discovery to automatically generate new state-of-the-art preference optimization algorithms.

## 4.1 Discovery Task - Multi-turn Dialogue on MT-Bench

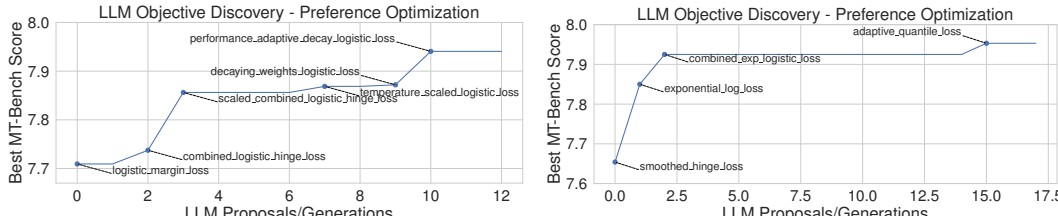

Figure 3: Examples of LLM Objective Discovery improvement across generations. The first and second runs are shown left and right respectively.

In this section we use our LLM-driven discovery method to discover new objective functions $f$ for offline preference optimization, as defined in Section 2 and Equation (3). Specifically, at each generation $i$, GPT-4 generates PyTorch [Paszke et al., 2017] code of candidate objective function $f_i$. Each objective function takes as input the variables of $\{\log \pi_\theta(y_w|x), \log \pi_{\text{ref}}(y_w|x), \log \pi_\theta(y_l|x), \log \pi_{\text{ref}}(y_l|x)\}$, and returns a scalar. For each proposed objective $f_i$, we check if $f_i$ is valid with a unit test.

For each *valid* generated objective function $f_i$, we finetune an LLM and then collect a performance evaluation score. Specifically, we build on top of the 'alignment-handbook' [Tunstall et al., 2023a] repository to finetune our models. Notably, this repository, when using DPO, reproduces 'Zephyr 7B Gemma'[2] Tunstall and Schmid [2024], Tunstall et al. [2023b], which at the time of release, achieved state-of-the-art scores on MT-Bench for 7B models. 'Zephyr 7B Gemma' first takes gemma-7b [Gemma-Team et al., 2024] and finetunes it on the 'deita-10k-v0-sft' dataset [Liu et al., 2023] to produce 'zephyr-7b-gemma-sft'[3]. It is then trained on the pairwise preference dataset of 'Argilla DPO Mix 7K'[4]. When evaluating a new objective function, we replace DPO in this last step with the generated objective function, keeping the same hyperparameters. We show example runs in Figure 12 and provide further experimental details in Appendix B.

Once we have a trained LLM for the proposed objective function $f_i$, we evaluate that LLM on the popular multi-turn dialogue evaluation benchmark of MT-Bench [Zheng et al., 2024]. This is a multi-turn open-ended question set, which uses GPT-4 to assess the quality of the trained model's responses, obtaining a high correlation with the popular Chatbot Arena [Zheng et al., 2024]. We provide further evaluation details in Appendix C.

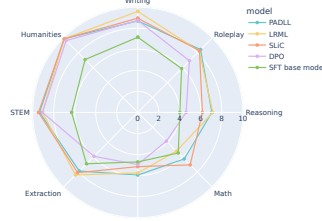

## 4.2 Discovery Results

Figure 4: MT-Bench Discovered Objective Evaluations

After evaluating approximately 100 objective functions, we catalogued the best-performing ones in Table 1. We tabulate the high-level objective forms here and provide the full objective loss functions and their associated code in Appendix E. Moreover, we also plot the best performing sub-task evaluations in Figure 4.

Table 1: **Discovery Task MT-Bench Evaluation Scores for each discovered objective function $f$.** We provide the baselines first, followed by a dashed line to separate the objective functions that were discovered. We provide details for each objective function in Appendix E.

| Name | Full Name | Objective $f$ Function | Score (/ 10) ↑ |
|------|-----------|------------------------|----------------|
| DPO | Direct Preference Optimization | $\log(1 + \exp(-\beta\rho))$ | 7.888 |
| DPO* | Official HuggingFace 'zephyr-7b-gemma' DPO model | $\log(1 + \exp(-\beta\rho))$ | 7.810 |
| SLiC | Sequence Likelihood Calibration | $\text{ReLU}(1 - \beta\rho)$ | 7.881 |
| KTO | Pairwise Kahneman-Tversky Optimization | see [Ethayarajh et al., 2024] | 7.603 |
| DBAQL | Dynamic Blended Adaptive Quantile Loss | $\sigma(\text{Var}[\beta\rho/\tau]) \cdot f_{dpo}(\beta\rho/0.9) + (1 - \sigma(\text{Var}[\beta\rho/\tau])) \cdot f_{exp}(\beta\rho \cdot 0.9)$ | **7.978** |
| AQL | Adaptive Quantile Loss | $q \cdot f_{dpo}(\beta\rho) + (1 - q) \cdot f_{slic}(\beta\rho)$ | 7.953 |
| PADLL | Performance Adaptive Decay Logistic Loss | $0.9 \cdot (1 - 0.5 \cdot \mathbb{1}[\rho < 0]) \cdot f_{dpo}(\beta\rho)$ | 7.941 |
| AQFL | Adaptive Quantile Feedback Loss | $r \cdot f_{dpo}(\beta\rho) + (1 - r) \cdot f_{slic}(\beta\rho)$ | 7.931 |
| CELL | Combined Exponential + Logistic Loss | $0.5 \cdot f_{dpo}(\beta\rho) + 0.5 \cdot f_{exp}(\beta\rho)$ | 7.925 |
| LRML (**DiscoPOP**) | Log Ratio Modulated Loss | $(1 - \sigma(\beta\rho/\tau)) \cdot f_{dpo}(\beta\rho) + \sigma(\beta\rho/\tau) \cdot f_{exp}(\beta\rho)$ | 7.916 |
| PFL | Policy Focused Loss | $1/2 \cdot f_{dpo}(\beta\rho) \cdot \mathbb{1}[\pi_w > \pi_r] + 2 \cdot f_{slic}(\beta\rho) \cdot \mathbb{1}[\pi_w \leq \pi_r]$ | 7.900 |

[2] https://huggingface.co/HuggingFaceH4/zephyr-7b-gemma-v0.1

[3] https://huggingface.co/HuggingFaceH4/zephyr-7b-gemma-sft-v0.1

[4] https://huggingface.co/datasets/argilla/dpo-mix-7k

# 5 Held-Out Evaluations

We next validate each of our discovered objective functions (shown in Table 1) on held-out tasks. We find that the Performance Adaptive Decay Loss (PADLL) and the Log Ratio Modulated Loss (LRML) consistently perform well. Because of its unconventional properties and performance, we refer to LRML as our discovered preference optimization, or *DiscoPOP*, algorithm.

We consider three different standard [Rafailov et al., 2023] open-ended text generation tasks each designed to evaluate different properties of the fine-tuned LLM policy $\pi_\theta$ where each LLM policy is trained with one of our discovered objective functions $f$ on a preference dataset $\mathcal{D} = \{(x^i, y_w^i, y_l^i)\}_{i=1}^N$.

## 5.1 Single-turn Dialogue - Alpaca Eval 2.0

We evaluate the trained models on Alpaca Eval 2.0, [Li et al., 2023, Dubois et al., 2023, 2024]. This is a single-turn dialogue LLM-based automatic evaluation using GPT-4 to assess the win rate of the trained LLM policy's completion compared to the of the underlying SFT base model. Alpaca Eval 2.0[5], has been validated against 20K human annotations, and aims to reduce the length bias of Alpaca Eval 1.0; where using length controlled (LC) Alpaca Eval shows a correlation with Chatbot Area of 0.98, making it a popular benchmark with the highest correlation to Chatbot Arena [Dubois et al., 2024]. We also detail task training details in Appendix B.1.

Table 2: **Alpaca Eval 2.0 - Held Out Single Turn Dialogue Task**. Win rate of the discovered objective functions $f$ evaluated on the Alpaca Eval 2.0 task against either GPT-4 or the SFT base model. Some of the discovered objective functions outperform the baselines, with the best bolded. We detail evaluation and error bars in Appendix C. We have highlighted the best scores with overlapping the standard errors.

| Function | Win Rate (%) ↑ | Win Rate - LC (%) ↑ | Win Rate (%) ↑ | Win Rate - LC (%) ↑ |
|---|---|---|---|---|
| | vs. GPT-4 | | vs. SFT Checkpoint | |
| DPO | $11.23 \pm 0.97$ | $12.81 \pm 0.66$ | $78.72 \pm 1.26$ | $63.34 \pm 0.30$ |
| DPO* | $11.99 \pm 1.00$ | $14.73 \pm 0.71$ | $75.75 \pm 1.31$ | $59.88 \pm 0.41$ |
| SLiC | $10.67 \pm 0.94$ | $13;16 \pm 0.69$ | $75.05 \pm 1.34$ | $59.67 \pm 0.42$ |
| KTO | $12.57 \pm 1.00$ | $13.58 \pm 0.67$ | $78.81 \pm 1.25$ | $62.76 \pm 0.31$ |
| DBAQL | $10.68 \pm 0.92$ | $11.41 \pm 0.57$ | $72.06 \pm 1.42$ | $54.40 \pm 0.38$ |
| AQL | $11.11 \pm 0.96$ | $13.63 \pm 0.68$ | $76.34 \pm 1.30$ | $60.94 \pm 0.36$ |
| PADLL | $\mathbf{14.07 \pm 1.04}$ | $\mathbf{14.89 \pm 0.66}$ | $\mathbf{81.10 \pm 1.21}$ | $64.14 \pm 0.28$ |
| AQFL | $\mathbf{13.63 \pm 1.05}$ | $\mathbf{15.55 \pm 0.71}$ | $79.32 \pm 1.23$ | $64.41 \pm 0.34$ |
| CELL | $10.27 \pm 0.93$ | $12.26 \pm 0.61$ | $71.75 \pm 1.39$ | $57.48 \pm 0.34$ |
| LRML | $\mathbf{13.21 \pm 1.02}$ | $\mathbf{14.78 \pm 0.67}$ | $\mathbf{79.27 \pm 1.24}$ | $\mathbf{65.18 \pm 0.32}$ |
| PFL | $8.15 \pm 0.83$ | $10.67 \pm 0.57$ | $68.27 \pm 1.44$ | $56.14 \pm 0.43$ |

We provide the Alpaca Eval 2.0 results in Table 2. As reference policies, we used GPT-4 for absolute comparison and the SFT-trained model for relative comparison. We observe that the discovered LRML (DiscoPOP), PADLL, and AQFL functions outperform the baselines and other discovered losses on the normal and length-controlled win rates. The differences in scores among these top-performing losses are not significant, except for the LC win rate against the SFT reference model, where DiscoPOP performs best.

## 5.2 Summarization (TL;DR)

We train an LLM policy to, given a forum post on Reddit $x$, generate a summarization $y$ of the main points. We finetune 'zephyr-7b-gemma-sft' using 10% of the Reddit TL;DR summarization preference dataset [Völske et al., 2017] on each of the baseline and discovered objective functions. As a reference model, we again use 'zephyr-7b-gemma-sft'. Further details on the training pipeline are outlined in Appendix B.2. To evaluate the quality of the summaries, we make use of the Alpaca Eval 2.0 library with a custom evaluation dataset existing of 694 test samples from the TL;DR dataset and a custom GPT-4 annotator template as described in Rafailov et al. [2023]. For additional details regarding the summarization evaluation see Appendix C.3.

---

[5] https://github.com/tatsu-lab/alpaca_eval

In Table 3 the PADLL loss and DPO loss perform best, with little difference from each other, on the summarization task in three out of four metrics. Additionally, the LRML - DiscoPOP function achieves scores slightly below the top performers, especially in the length-controlled win rates. In contrast to the single-turn dialogue task, the AQFL loss does not achieve high scores in the held-out evaluation.

Table 3: **TL;DR - Held Out Summarization Task** Win-rate of various preference optimization functions in the summarization task was evaluated with the Alpaca Eval 2.0 calculations, against a subset of the test set (694 samples). The baseline outputs are the human-generated preferences, and the model after SFT (see Appendix C for details). Note that the standard error in the LC win-rate has been rounded down because of values $< 0.001$. We have highlighted the scores with means overlapping the standard error of the best score.

| Function | Win Rate (%) ↑ | Win Rate - LC (%) ↑ | Win Rate (%) ↑ | Win Rate - LC (%) ↑ |
| --- | --- | --- | --- | --- |
| | vs. Human Preference | | vs. SFT Checkpoint | |
| DPO | $\mathbf{88.27 \pm 1.07}$ | $\mathbf{82.82 \pm 0.00}$ | $\mathbf{54.38 \pm 1.52}$ | $54.64 \pm 0.00$ |
| SLiC | $83.02 \pm 1.29$ | $63.41 \pm 0.00$ | $53.03 \pm 1.52$ | $54.11 \pm 0.00$ |
| KTO | $85.34 \pm 1.18$ | $80.26 \pm 0.00$ | $51.15 \pm 1.54$ | $50.0 \pm 0.00$ |
| DBAQL | $84.71 \pm 1.21$ | $78.68 \pm 0.00$ | $52.55 \pm 1.52$ | $55.14 \pm 0.00$ |
| AQL | $81.87 \pm 1.32$ | $68.89 \pm 0.00$ | $46.00 \pm 1.54$ | $50.0 \pm 0.00$ |
| PADLL | $\mathbf{88.54 \pm 1.05}$ | $76.13 \pm 0.00$ | $\mathbf{55.34 \pm 1.52}$ | $\mathbf{55.64 \pm 0.00}$ |
| AQFL | $85.03 \pm 1.22$ | $76.23 \pm 0.00$ | $49.56 \pm 1.53$ | $50.38 \pm 0.00$ |
| CELL | $86.33 \pm 1.14$ | $73.72 \pm 0.00$ | $50.35 \pm 1.52$ | $51.90 \pm 0.00$ |
| LRML | $\mathbf{87.63 \pm 1.10}$ | $\underline{81.88 \pm 0.00}$ | $\underline{53.46 \pm 1.52}$ | $55.10 \pm 0.00$ |
| PFL | $79.84 \pm 1.35$ | $69.23 \pm 0.00$ | $44.12 \pm 1.52$ | $44.57 \pm 0.00$ |

## 5.3 Positive sentiment generation (IMDb)

In this task, we train an LLM policy to generate movie review completions $y$ with positive sentiment, where $x$ is a prompt at the start of a movie review from the IMDb dataset [Maas et al., 2011]. We start with a GPT-2 [Radford et al., 2019] model, which had supervised fine-tuning on the IMDb dataset, and we perform preference optimization using the baseline and discovered objective loss functions. Details of the training implementations can be found in Appendix B.3. Inspired by Rafailov et al. [2023]'s experiments, we calculate the model rewards through a pre-trained sentiment classifier, which we use as a proxy for ground truth, as well as the KL-Divergence of the trained model and the reference model. Appendix C.4 provides further details into the evaluation for this task.

We provide results of models with converging $\beta$ values in Figure 5 for LRML compared against DPO and SLiC, displaying the model rewards against the KL-Divergence to the reference model. In Figure 5a, the LRML-trained text generator outperforms the DPO model in terms of rewards and KL-divergence with low $\beta$ values (0.025, 0.05, 0.1). At higher $\beta$ values (0.5 and 1.0) both methods show trends of increased KL-Divergence and lower rewards, but generally, LRML maintains a higher reward than DPO. In Figure 5b, we note that LRML slightly outperforms DPO, SLiC, AQFL, and PADLL at $\beta \in \{0.05, 0.1\}$ in terms of reward. For larger $\beta$ values (0.5 and 1.0), LRML shows similar trends of increased KL-Divergence and rewards like the other objective functions. A more detailed comparison between the individual discovered losses and the baselines can be found in Appendix Figure 8.

## 6 Analysis of DiscoPOP

We list all our discovered objectives in Table 1, as well as the code and mathematical representations in Appendix E. In this section, we now analyze the Log Ratio Modulated Loss, which we define as the DiscoPOP loss function, as it performs consistently high across the held-out evaluation tasks, and we provide some intuitive understanding of how it outperforms the existing state-of-the-art objectives.

### 6.1 Log Ratio Modulated Loss (DiscoPOP)

The Log Ratio Modulated Loss is a dynamically weighted sum of the logistic loss (as used in DPO) and the exponential loss. The weight of each is determined through a sigmoid calculation of the

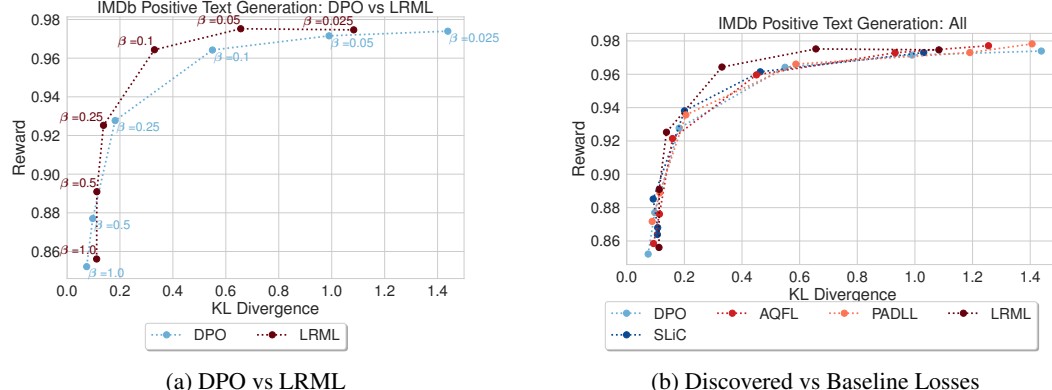

(a) DPO vs LRML

(b) Discovered vs Baseline Losses

Figure 5: Frontiers of expected reward vs KL divergence for converging models for the LRML against DPO and SLiC objective function. The rewards and KL-divergence values are averaged over 10 generations with different seeds. The sweep is done over $\beta \in \{0.025, 0.05, 0.1, 0.25, 0.5, 1.0\}$. The optimal point is the top left corner, where the perfect reward is achieved with minimal divergence from the reference model.

difference of log-ratios ($\rho$). Mathematically, the LRML function can be described with a temperature parameter $\tau = 0.05$ as follows:

$$f_{lrml}(\beta\rho) = (\sigma(\beta\rho/\tau) - 1) \cdot f_{dpo}(\beta\rho) + \sigma(\beta\rho/\tau) \cdot f_{exp}(\beta\rho) \tag{4}$$

$$= (1 - \sigma(\beta\rho/\tau)) \cdot \log(1 + \exp(-\beta\rho)) + \sigma(\beta\rho/\tau) \cdot \exp(-\beta\rho) \tag{5}$$

If the difference of log ratios is zero ($\rho = 0$), which is at the start of the training when the model policy $\pi_\theta$ is equal to the reference policy $\pi_{\text{ref}}$, then the loss is equally balanced between the logistic and exponential loss. If $\rho \to \infty$, the model policy diverges from the reference policy and chosen outputs are preferred, then the exponential term dominates. This emphasizes larger differences more strongly. On the other hand, if $\rho \to -\infty$, the model policy diverges from the reference policy, and rejected outputs are preferred. In this case, the logistic loss can handle moderate differences well. The baseline objective losses and the LRML, the PADLL, and the AQFL functions are displayed in Figure 6, including their gradients. Surprisingly, the DiscoPOP function has a non-convex segment and negative gradients at the starting point $\rho = 0$. This is potentially helpful for introducing a curriculum or for stochasticity. Additional results and analysis of the discovered loss function can be found in Appendix D.6.

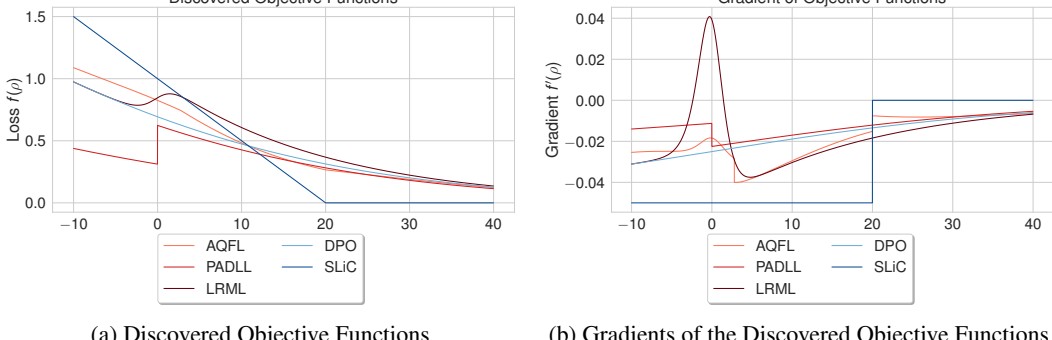

(a) Discovered Objective Functions

(b) Gradients of the Discovered Objective Functions

Figure 6: Figure 6a: Baseline objective functions DPO and SLiC, and the discovered ones, LRML, AQFL, and PADLL. Figure 6b: gradients of the objectives as a function of $\rho$ and with fixed $\beta = 0.05$.

## 6.2 Limitations of DiscoPOP

While performing very well on single-turn text generation and text summarization, we observed during the IMDb experiment that LRML struggles to converge when $\beta$ is too low ($\beta \leq 0.01$) or too high ($\beta \geq 2.5$), likely because $\beta \neq 0.05$ was never seen or used during the discovery process.

In Figure 9 and Figure 10 of the Appendix, we plot the LRML objective function for $\beta \in \{0.01, 0.025, 0.05, 0.1, 0.25, 0.5, 1, 2.5, 5\}$ against DPO. When $\beta$ is high, the DiscoPOP objective function takes the form of the DPO log sigmoid loss. During training on $\beta = 0.01$, we observed that DiscoPOP gets stuck in generating predominantly negative reviews (resulting in a reward score of $\sim 0.15$). We hypothesize that the loss is stuck in the local minima to the left with a negative difference in log ratios. While training with $\beta \in \{2.5, 5.0\}$, we observed that the model collapsed after a sharp spike in the loss and subsequently having loss value 0 and NaN outputs. This is potentially due to a large gradient in the non-convex part, which could be amended with gradient clipping.

## 7 Related Work

**Evolution and Search with Large Language Models**. LLMs provide a fast and automated way to create multiple candidate solutions for a problem stated in natural language [Song et al., 2024], which makes them powerful tools for driving population-based search procedures. Various recent works have applied this approach to coding problems [Romera-Paredes et al., 2024], neural architecture search [Chen et al., 2024a, Holt et al., 2024b], virtual robotic design settings [Lehman et al., 2023], reward functions [Ma et al., 2023, Yu et al., 2023], and algorithm heuristics [Liu et al., 2024]. Finally, recently LLMs have shown to be capable of acting as recombination operators for black-box optimization with Evolution Strategies [Lange et al., 2024] and for Quality-Diversity approaches [Lim et al., 2024].

**Automated Discovery for Machine Learning**. There are many other approaches to automating the discovery of generalizable machine learning algorithms. Some prior works explore the space of ML functions using genetic algorithms and a hand-crafted domain-specific language for reinforcement learning algorithms [Co-Reyes et al., 2021], curiosity algorithms [Alet et al., 2020], and optimizers [Chen et al., 2024b]. Other works instead parameterize a transferrable objective function using neural networks and optimize them with evolution strategies or meta-gradients. For example, Lu et al. [2022], Jackson et al. [2024b], Houthooft et al. [2018], Alfano et al. [2024], Kirsch et al. [2019], Oh et al. [2020], Jackson et al. [2024a] discover policy optimization objectives, Metz et al. [2022] evolves neural network optimizers, and Lange et al. [2023b,a] evolve blackbox optimizers. Moreover, automatically discovering closed-form functions (i.e., symbolic regression), works exist that leverage RL [Petersen et al., 2020], gradient descent [Kacprzyk et al., 2024], RL with evolution strategies [Mundhenk et al., 2021], pre-training transformers [Biggio et al., 2021] and hybrid combinations of pre-training transformers, which are further refined with RL and evolution strategies [Holt et al., 2023b].

**Preference Optimization Algorithms**. While the reduction to supervised learning makes DPO and alternatives easier to use, other approaches have sought to simplify the RL step, including using variants of REINFORCE [Ahmadian et al., 2024, Gemma-Team et al., 2024] as well as more fine-grained feedback [Wu et al., 2024] through preferences over individual steps in the reasoning process [Uesato et al., 2022, Lightman et al., 2023] or reward redistribution [Chan et al., 2024]. Others use iterative offline training interleaved with sampling from the policy model and obtaining a preference ranking from themselves [Xu et al., 2023], another judge LLM [Guo et al., 2024], or an oracle [Swamy et al., 2024].

## 8 Conclusion

**Summary**. In this paper, we proposed and used LLM-driven objective discovery to generate novel offline preference optimization algorithms. Specifically, we were able to discover high-performing preference optimization losses that achieve strong performance across held-out evaluation tasks, with the highest performing providing new insights into what an optimal objective may need to possess, such as being a blend of logistic and exponential losses and possibly being non-convex.

**Limitations & Future work**. There are multiple limitations to our current approach. First, we have only scratched the surface of how to generate LLM objective proposals effectively. Initial exploratory experiments using techniques such as temperature sampling or worst-to-best performance sorting in the context did not yield significant improvements. But one could imagine leveraging more information about the training runs and automatically tuning instruction prompt templates. E.g. by providing entire learning curve plots to a Visual Language Model (see Figure 13) or by meta-meta-optimizing [Lu et al., 2023] the LLM prompt. Second, the highest-performing loss re-purposed $\beta$

in the traditional sense, making it affect the functional behavior and the KL penalty of the model with respect to the base model. This motivates future work to study different forms, with perhaps multiple floating point parameters in the form, that each could be tuned separately. Although we provided an initial analysis sweep over this one single parameter and observed some instances of the functional behavior leading to instability of training the model, a further multi-parameter analysis, reformulating the objective, would be beneficial for future work. Finally, our work uses closed-source models (GPT-4) to generate code, which limits reproducibility and is costly to run. Future work could use the produced models *themselves* to generate code, resulting in code-level self-improvement.

**Broader Impact and Ethical Considerations**. This paper presents an LLM-driven discovery in-context learning pipeline that is used to generate better-performing novel offline preference optimization algorithms. However, misuse of the pipeline as a tool or training an LLM to produce undesirable, unethical, or harmful outputs could be possible by a user. Furthermore, due to the use of LLMs and training of LLMs, the outputs are susceptible to hallucinations, motivating all outputs of the LLMs to always have a content filter applied to the outputs. Finally, this work takes a small step towards code-level self-improvement in language models, which could potentially result in unintended behaviors.

## Acknowledgments and Disclosure of Funding

This work was supported by Azure sponsorship credits granted by Microsoft's AI for Good Research Lab and by Microsoft's Accelerate Foundation Models Academic Research initiative. The hardware used for training was sponsored by GoodAI. SH is funded by AstraZeneca. AJC is funded by a Microsoft Research and EPSRC ICASE scholarship award. CL and RTL were supported by Sakana AI at the time of this work. The code can also be accessed at https://github.com/samholt/DiscoPOP.

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

# Appendix

## Table of Contents

## A   LLM-Driven Objective Discovery Implementation Details

### A.1   Prompts

We use the following system prompt to generate the model responses:

```
You are a machine learning researcher who is testing out different
    RLHF loss functions. When you respond, output a JSON where the
    first key ("thought") corresponds to your thought process when
    designing the next function. The second key ("name") corresponds
    to the name of your next function. Finally, the last key ("code")
    corresponds to the exact python code that you would like to try.
    Here is an example:

{
"thought": "Based on the previous outputs, I should try the direct
    preference optimization algorithm.",
"name": "dpo",
"code": "def sigmoid_loss(
    self,
    policy_chosen_logps: torch.FloatTensor,
    policy_rejected_logps: torch.FloatTensor,
    reference_chosen_logps: torch.FloatTensor,
    reference_rejected_logps: torch.FloatTensor,
) -> torch.FloatTensor:
    pi_logratios = policy_chosen_logps - policy_rejected_logps
    ref_logratios = reference_chosen_logps - reference_rejected_logps
    logits = pi_logratios - ref_logratios
    losses = -F.logsigmoid(self.beta * logits)
    return losses"
}

You are deeply familiar with binary classification losses from the
    literature. Be creative and reference prior literature when
    possible.

You must use the exact function interface used above. Feel free to
    define extra hyperparameters within your function as constants. Do
     not make them attributes of self.

Note that 'self.beta = 0.05'.

RLHF loss functions train on a dataset of pairs of preferred and
    rejected completions.
'policy_chosen_logps' refers to the policy's log probabilities of the
    preferred completion, and 'policy_rejected_logps' refers to the
    policy's log probabilities of the rejected completion.
'reference_chosen_logps' and 'reference_rejected_logps' refer to the
    same for the reference (base) model.

The user will then return to you a fitness that corresponds to the
    performance of the resulting model on a downstream task. Your goal
     is to maximize performance.
```

We then provide the first user prompt as such:

```
Here are some results we've obtained:

[
{
    "code": "
def logistic_log_loss(
    self,
    policy_chosen_logps: torch.FloatTensor,
```

```
    policy_rejected_logps: torch.FloatTensor,
    reference_chosen_logps: torch.FloatTensor,
    reference_rejected_logps: torch.FloatTensor,
) -> torch.FloatTensor:
    pi_logratios = policy_chosen_logps - policy_rejected_logps
    ref_logratios = reference_chosen_logps - reference_rejected_logps
    logits = pi_logratios - ref_logratios
    losses = -F.logsigmoid(self.beta * logits)
    return losses
    ",
    "fitness": 7.8875
},
{
    "code": "
def hinge_loss(
    self,
    policy_chosen_logps: torch.FloatTensor,
    policy_rejected_logps: torch.FloatTensor,
    reference_chosen_logps: torch.FloatTensor,
    reference_rejected_logps: torch.FloatTensor,
) -> torch.FloatTensor:
    pi_logratios = policy_chosen_logps - policy_rejected_logps
    ref_logratios = reference_chosen_logps - reference_rejected_logps
    logits = pi_logratios - ref_logratios
    losses = torch.relu(1 - self.beta * logits)
    return losses
    ",
    "fitness": 7.88125
},
{
    "code": "
def ipo_loss(
    self,
    policy_chosen_logps: torch.FloatTensor,
    policy_rejected_logps: torch.FloatTensor,
    reference_chosen_logps: torch.FloatTensor,
    reference_rejected_logps: torch.FloatTensor,
) -> torch.FloatTensor:
    pi_logratios = policy_chosen_logps - policy_rejected_logps
    ref_logratios = reference_chosen_logps - reference_rejected_logps
    logits = pi_logratios - ref_logratios
    losses = (logits - 1 / (2 * self.beta)) ** 2
    return losses
    ",
    "fitness": 7.84
},
{
    "code": "
def kto_pair_loss(
    self,
    policy_chosen_logps: torch.FloatTensor,
    policy_rejected_logps: torch.FloatTensor,
    reference_chosen_logps: torch.FloatTensor,
    reference_rejected_logps: torch.FloatTensor,
) -> torch.FloatTensor:
    chosen_KL = (policy_chosen_logps - reference_chosen_logps).mean().
    clamp(min=0)
    rejected_KL = (policy_rejected_logps - reference_rejected_logps).
    mean().clamp(min=0)

    chosen_logratios = policy_chosen_logps - reference_chosen_logps
    rejected_logratios = policy_rejected_logps -
    reference_rejected_logps
    # As described in the KTO report, the KL term for chosen (rejected
    ) is estimated using the rejected (chosen) half.
```

```
    losses = torch.cat(
        (
            1 - F.sigmoid(self.beta * (chosen_logratios - rejected_KL)
    ),
            1 - F.sigmoid(self.beta * (chosen_KL - rejected_logratios)
    ),
        ),
        0,
    )
    return losses
    ",
    "fitness": 7.603125
}
]

Please generate the next one.
```

Upon testing the generated code, if an error is encountered, we provide the following prompt, where 'error' is the text containing the system error:

```
Code not valid. Error:
{error}
Please generate the next one.
```

Upon successful completion, we return the following user prompt, where 'val' is the MT-Bench score:

```
Fitness: {val}.
Please generate the next one.
```

# B   Training Details

## B.1   Discovery Task - Single-turn Dialogue

For each valid generated objective function $f_i$, we use it to train an LLM and then collect a performance evaluation score. Specifically, we follow the same process when training and evaluating all objective functions, starting with a pre-trained supervised fine-tuned (SFT) 7 billion gemma model of 'zephyr-7b-gemma-sft' This is a 7 billion base version gemma [Gemma-Team et al., 2024] model supervised-fine-tuned on the 'deita-10k-v0-sft' dataset [Liu et al., 2023]. Starting with this model, we train it on the pairwise preference dataset of 'Argilla DPO Mix 7K'; which attempts to create a high-quality preference dataset by filtering only highly rated chosen responses from the datasets of a multi-turn dataset, instruction following dataset [Longpre et al., 2023] and a diverse preference dataset that covers truthfulness, honesty and helpfulness [Cui et al., 2023]. For each training run, we trained all the parameters of the starting model, using a fixed $\beta = 0.05$. We used the same fixed hyper-parameters for all training runs unless explicitly noted. Specifically, we used a learning rate of 5e-7, bfloat16 floating-point format, two epochs, a batch size per device of two, a gradient accumulation step of 8, a cosine learning rate scheduler, and AdamW optimization algorithm [Loshchilov and Hutter, 2017]. We use the popular TRL transformers library [von Werra et al.], adapting the offline preference optimization objective function to train all models. The models were trained on 8 Nvidia A100 GPUs. An individual training run takes approximately 30 minutes. We provide training and evaluation statistics for discovered objective functions in Figure 7. We also provide the equivalent code implementation at https://github.com/vanderschaarlab/DiscoPOP.

## B.2   TL;DR Summarization

To determine if the discovered objective functions generalize well also to other tasks, we use them to preference optimize an LLM for text summarization. Specifically, we start again with a pre-trained supervised fine-tuned (SFT) 7 billion gemma model of 'zephyr-7b-gemma-sft', and we optimized it with the objective function $f_i$ on a subsample of the Reddit TL;DR summarization preference dataset

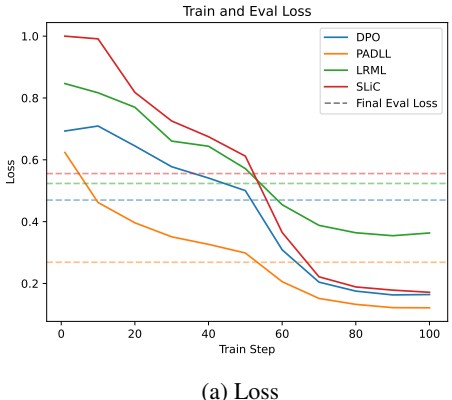
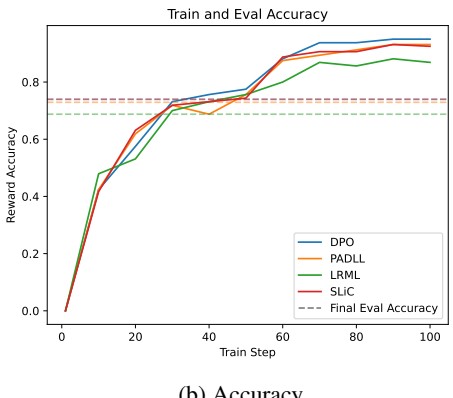

|          |              |
|:--------:|:------------:|
| (a) Loss | (b) Accuracy |

Figure 7: Training and eval statistics of DPO, SLiC, PADLL, and LRML. The losses are not directly comparable to each other, as they are calculated differently for each model. Interestingly, eval results are not strongly correlated with the downstream MT-Bench scores, as LRML achieves the worst accuracy.

[Völske et al., 2017][6]. More precisely we use the first 10% of the dataset for preference optimization, which amounts to around 8'000 training samples. During training the hyperparameters are kept the same as in the single-turn dialogue task, explained in subsection B.1, except that LLMs were trained 4 Nvidia A100 GPUS using a gradient accumulation step of 16. An individual training run takes approximately 1.5 hours.

## B.3 IMDb Positive Text Generation

Another popular generalization task for preference optimization [Rafailov et al., 2023] is to fine-tune a small LLM to generate positive text for movie review, based on the IMDb sentiment dataset [Maas et al., 2011][7] following the the work of Zhou et al. [2024]. As a starting model, we use a GPT2 model [Radford et al., 2019], that was supervised fine-tuned on the IMDb dataset[8]. Subsequently, we apply the baseline and discovered objective function $f_i$ for preference optimization. The goal of the LLM is to give a short prompt of 2-8 tokens, which indicate the start of a movie review, to generate a positive review. As we are interested in the effect of $\beta$ on the rewards and KL-Divergence, we train the objective functions over a sweep of $\beta \in \{0.01, 0.025, 0.05, 0.1, 0.25, 0.5, 1, 2.5, 5\}$. Every LLM is trained for three epochs, using the AdamW optimizer, with an initial learning rate of 5.0e-5, a warm-up scheduler of 0.1, and a cosine learning rate scheduler. The models are trained on 4 Nvidia A100 GPUs, using a gradient accumulation step of 8, and a batch size per device of 2. The training takes around 30 minutes.

## C Evaluation Metrics

### C.1 MT-Bench

To assess the fitness of the discovered preference optimization loss function during the discovery phase, we evaluate the trained LLMs on the MT-Bench [Zheng et al., 2024] benchmark. The evaluation benchmark consists of 80 high-quality multi-turn questions from various disciplines. The goal is to assess LLM's ability to follow instructions and keep the flow of a conversation. A larger LLM, in our case GPT-4, is then used as a judge to score the quality of the answers with a number from 0 (lowest) to 10 (highest). Scores are given based on the quality of the LLM's first-turn answer (single-turn) and first and second answers (multi-turn). Finally, the MT-Bench score is the average

---

[6]https://huggingface.co/datasets/CarperAI/openai_summarize_comparisons
[7]https://huggingface.co/datasets/ZHZisZZ/imdb_preference
[8]https://huggingface.co/lvwerra/gpt2-imdb

of single-turn and multi-turn scores. For answer generation and evaluation, we used the FastChat library[9] and its standard sampling and temperature parameters, provided by Zheng et al. [2024].

## C.2    Alpaca Eval

Alpaca Eval 2.0 [Li et al., 2023, Dubois et al., 2023, 2024] is also a popular benchmark for evaluating LLMs. This is a single-turn dialogue LLM-based automatic evaluation using a stronger LLM, here GPT-4 Turbo, to assess the win rate of the trained LLM policy's completion compared to either GPT-4 or the of the underlying SFT base model. Specifically, Alpaca Eval 2.0 has been validated against 20K human annotations and aims to reduce the length bias of Alpaca Eval, where using length-controlled (LC) Alpaca Eval shows a correlation with Chatbot Arena of 0.98, making it a popular benchmark with the highest correlation to Chatbot Arena [Dubois et al., 2024]. The Alpaca evaluation dataset consists of 841 high-quality instructions from different data sets. The library[10] provided by Dubois et al. [2024] calculates the win-rate (percentage were the trained policy is prefered over the reference policy, first introduced in Alpaca Eval 1.0), and a length-controlled win-rate, where a linear model is fitted to de-bias for length of the prompt and instruction difficulty. We used a temperature of 0.7, sampling, and a maximum number of new tokens of 1024 to generate the answers. Furthermore, the library provides the standard error of the mean, which indicates the confidence of the win-rate and LC win-rate.

## C.3    TL;DR Summarization Win-Rate

To evaluate how well the discovered objective functions generalize to the summarization task, we use the Alpaca Eval 2.0 library, similar to subsection C.2. Instead of using the Alpaca evaluation dataset, we create a custom dataset of 694 samples from the IMDb preference test dataset. Additionally, we change the prompt of the annotator LLM, to fit the "Summarization GPT-4 win rate prompt (C)" as described in Rafailov et al. [2023]. The (LC) win-rate is calculated against either the existing human-chosen test sample or against the summary generated by the SFT reference model. For a summary generation, we apply a temperature parameter of 0.7, sampling, and a maximum of 256 new tokens. Moreover, we stop the summarization after the "\n" token to avoid nonsensical generations. Furthermore, as we cannot calculate an instruction difficulty for the length-controlled win-rate, we omit this term from the linear model (This has only a small impact on the metric). In addition to the win-rates we also provide the standard error as a measure of confidence.

## C.4    IMDb Rewards vs KL-Divergence

For the positive text generation, we do not require an LLM judge compared to MT-Bench, Alpaca Eval 2.0, and TL;DR evaluation, as we take a pre-trained sentiment classifier[11] as ground truth reward scorer. The LLMs apply sampling and a maximum of 60 new tokens for the positive text generation. The rewards and KL-divergence are averaged over 10 different generations from the trained LLMs.

---

[9]https://github.com/lm-sys/FastChat
[10]https://github.com/tatsu-lab/alpaca_eval
[11]https://huggingface.co/siebert/sentiment-roberta-large-english

# D    Additional Results

## D.1    Frontiers of Expected Reward vs KL Divergence

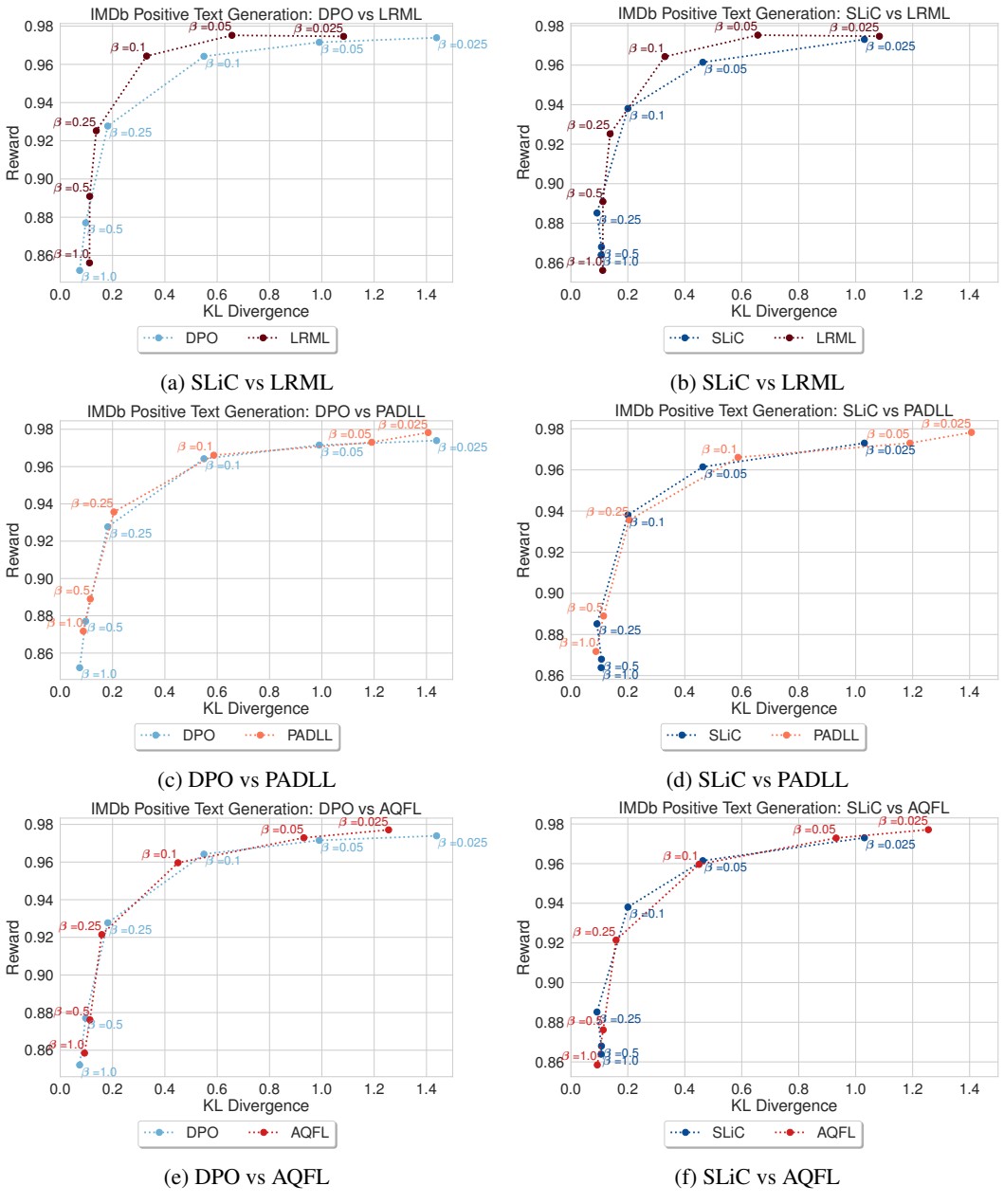

(a) SLiC vs LRML

(b) SLiC vs LRML

(c) DPO vs PADLL

(d) SLiC vs PADLL

(e) DPO vs AQFL

(f) SLiC vs AQFL

Figure 8: Frontiers of expected reward vs KL divergence after convergence for the baseline functions and all the discovered ones. The rewards and KL divergence values are averaged over 10 generations with different seeds. The sweep is done over $\beta \in \{0.025, 0.05, 0.1, 0.25, 0.5, 1, \}$. The optimal point is the top left corner, where perfect reward is achieved with minimal divergence from the reference model, to avoid reward hacking.

## D.2 Loss Sweeps for Different Beta Parameters

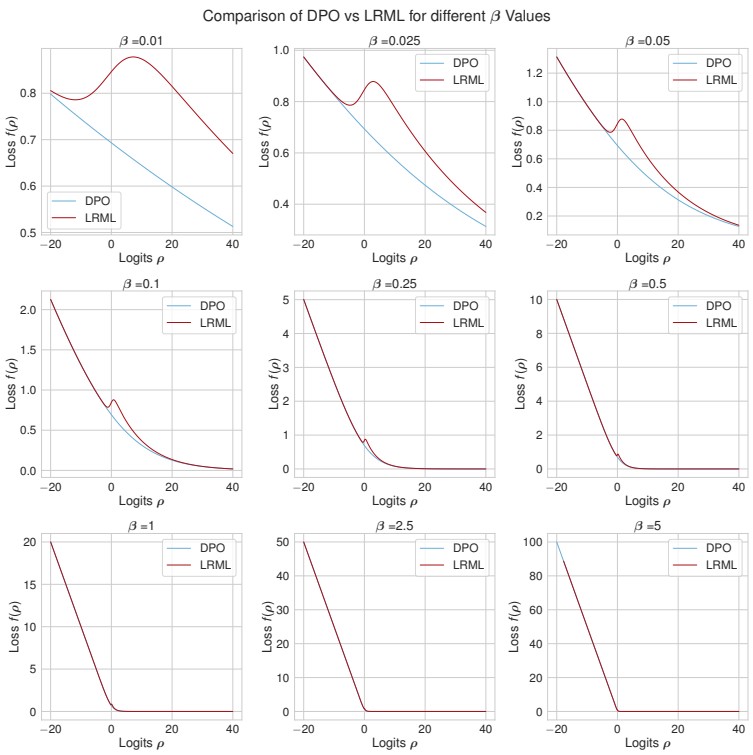

Figure 9: DPO and LRML objective function over $\beta \in \{0.01, 0.025, 0.05, 0.1, 0.25, 0.5, 1, 2.5, 5\}$.

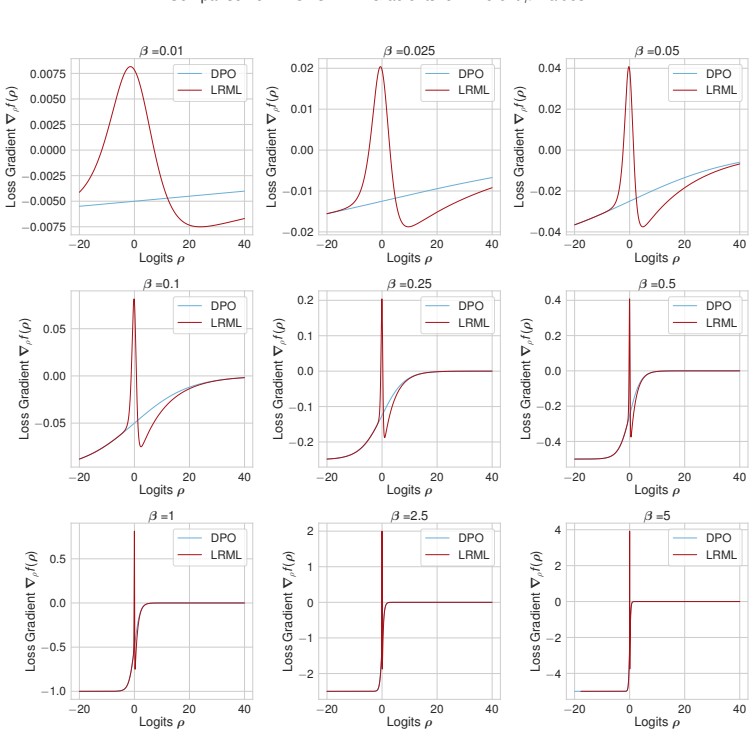

Figure 10: DPO and LRML gradient function over $\beta \in \{0.01, 0.025, 0.05, 0.1, 0.25, 0.5, 1, 2.5, 5\}$.

## D.3 Discovery Robustness with respect to LLM Hyperparameters

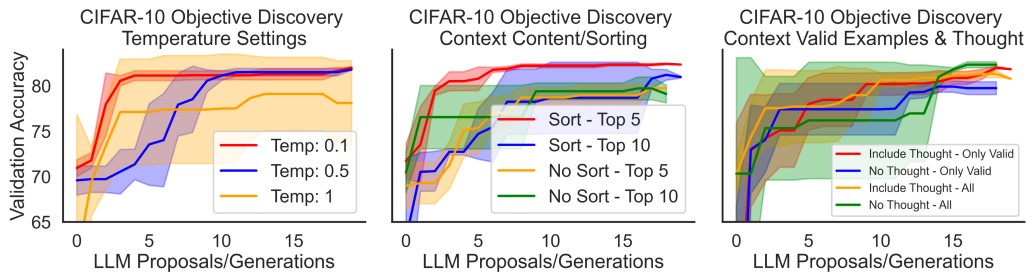

Figure 11: Robustness of the LLM-driven discovery process. **Left**. We compare different sampling temperatures $\{0.1, 0.5.1.0\}$. **Middle**. The default configuration includes all objective proposals and evaluations in chronological order. Here we also explore using only the top-$K$ performing objectives unsorted and sorted by their performance. **Right**. We also investigate whether using a "thought" as part of the context and whether to include non-valid code and error messages improves performance. The discovery process for CIFAR-10 objectives (5 epochs) is robust to these settings. The results are averaged across 3 independent runs.

## D.4 LLM-Driven Discovery Analysis

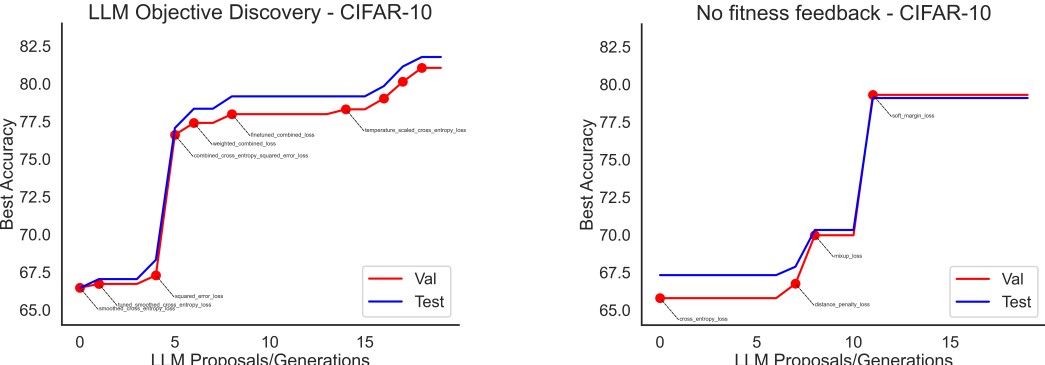

Figure 12: LLM-driven discovery for CIFAR-10 loss functions with (left) and without (right) providing fitness feedback. Note that without feedback, it performs worse but also is unable to *refine* its ideas, resulting in fewer improvements across generations.

## D.5 Visual Language Models for Objective Discovery

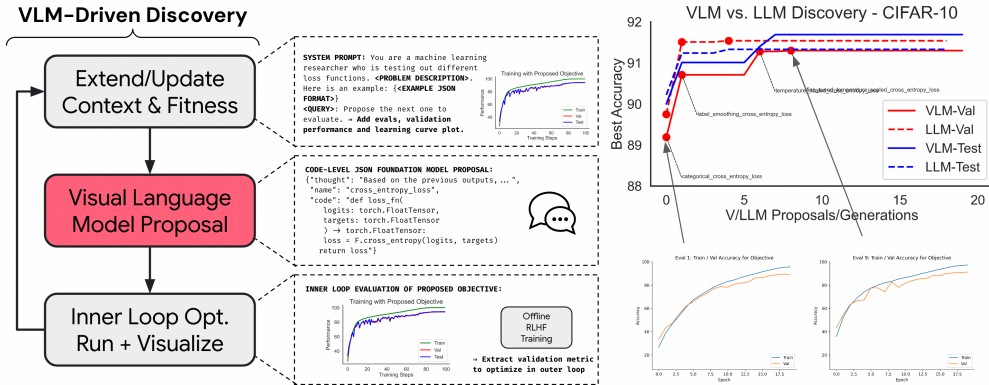

Figure 13: Objective Discovery with a Visual Language Model (VLM) for CIFAR-10 (20 epochs). We provide a plot of the training and validation accuracy across training as context components to the VLM (GPT-4-Turbo).

### D.6    Additional Analysis of DiscoPOP

We performed further analysis to improve our understanding of the discovered loss function. We hypothesize that the local optimum of the DiscoPOP loss could catch noisy or incorrect data points. By inspecting the DiscoPOP loss values and log ratios of the training preference pairs in the IMDb dataset, we see that 1.35% of training points fall there (see Figure 14). Although we use the binary preference labels from the IMDb dataset[12] for training, the dataset also includes a positivity reward score for each completion, calculated by a separate reward model. When we analyze the data points between the local optima, we find that the positive and negative completions are significantly closer in absolute reward difference than those outside the local optimum (See Table 4). This implies that the preference labels on those points are likely more challenging to distinguish and help empirically validate our hypothesis.

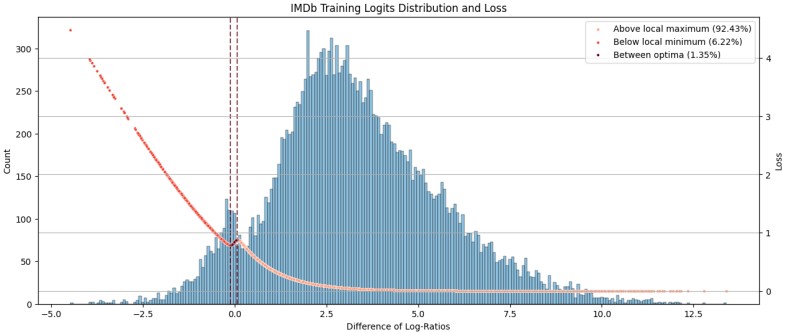

Figure 14: Distribution of $\beta$-scaled difference of log-ratios (left y-axis) and corresponding DiscoPOP loss value (right y-axis) of the training samples on the IMDb positive review generation task. The DiscoPOP function has a local minimum at $f_{lrml}(-2.3714) = 0.785929$ and a local maximum at $f_{lrml}(1.44012) = 0.87829$. The number of samples within the two local optima corresponds to 1.35% of the training set.

| Description | Domain $\beta\rho$ | Mean | 95%-CI | $p$-value | % of training set |
|---|---|---|---|---|---|
| Between optima | $[-2.3714, 1.44012]$ | 0.981 | (0.830, 1.131) | - | 1.35 |
| Full range | $\mathbb{R}$ | 3.861 | (3.818, 3.904) | $< 0.01\%$ | 100 |
| Outside optima | $\mathbb{R}\backslash[-2.3714, 1.44012]$ | 3.9 | (3.857, 3.944) | $< 0.01\%$ | 98.65 |
| Below local minimum | $(-\infty, -2.3714)$ | 4.086 | (4.041, 4.131) | $< 0.01\%$ | 6.22 |
| Above local maximum | $(1.44012, \infty)$ | 1.141 | (1.076, 1.206) | 4.29% | 92.43 |

Table 4: The IMDb positive review preference dataset also provides "golden reward" scores for the chosen responses $r_w$ and rejected responses $r_l$, calculated with a strong sentiment classifier $(\log p(\text{pos}) - \log p(\text{neg}))$. We analysed the absolute difference in reward scores between the chosen and rejected responses $|r_w - r_l|$ across the different domains of the DiscoPOP function and report the statistics here. Training samples within the optima have a significantly lower mean absolute difference in rewards compared to the other regions in the DiscoPOP loss function. The samples stuck within the optima are the training triplets where the chosen and rejected responses are "closest" to each other and likely the most "noisy" labels.

## E    Discovered Objective Functions

To describe the discovered losses mathematically, we define three existing preference optimization losses here:

$$f_{dpo}(\beta\rho) = -\log(\sigma(\beta\rho)) = -\log(\frac{1}{1 + \exp(-\beta\rho)}) = \log(1 + \exp(-\beta\rho)) \qquad (6)$$

$$f_{slic}(\beta\rho) = \text{ReLU}(1 - \beta\rho) \qquad (7)$$

[12]https://huggingface.co/datasets/ZHZisZZ/imdb_preference

$$f_{exp}(\beta\rho) = \exp(-\beta\rho) \tag{8}$$

Moreover, we display the code of the discovered losses as it is output by the LLM. In addition, we provide a mathematical representation of each, which we have adapted to be consistent with $\beta$ being the KL-Divergence regularization parameter. This is because the generated code for LRML, DBAQL, AQL, AQFL, and PFL did not uphold the $\beta$ ought to be multiplied with the difference of log-ratios before any further calculations. If this was not upheld, it could lead to the loss function changing shapes based on the KL-regularization term, and therefore, models could not converge or potentially collapse. In future work, we should constrain the exploring LLM to uphold the $\beta$ multiplication with the input before any other calculations are done with the difference of log-ratios $\rho$. As the meta-exploration was done with a set $\beta = 0.05$, and we wish to keep consistent with this scale of regularization, we have adapted the losses by dividing $\rho$ values used in intermediate calculations with a scalar $\tau = 0.05$.

In the IMDb experiment in Section 5, we have thus used the corrected version of codes for the discovered losses based on the provided mathematical representation, as we were most interested in the effect of the KL-divergence compared to the model rewards.

## E.1 DBAQL: Dynamic Blended Adaptive Quantile Loss

MT-Bench Score: 7.978

```python
def dynamic_blended_adaptive_quantile_loss(
    self,
    policy_chosen_logps: torch.FloatTensor,
    policy_rejected_logps: torch.FloatTensor,
    reference_chosen_logps: torch.FloatTensor,
    reference_rejected_logps: torch.FloatTensor,
) -> torch.FloatTensor:
    import torch.nn.functional as F
    # Constants for the loss function
    starting_quantile = 0.5
    quantile_adapt_rate = 0.01
    temperature = 0.9
    dynamic_blend_rate = 1.0

    pi_logratios = policy_chosen_logps - policy_rejected_logps
    ref_logratios = reference_chosen_logps - reference_rejected_logps
    logits = pi_logratios - ref_logratios
    logits_variability = logits.var()

    # Calculate an adaptive quantile based on a moving target
    moving_quantile = starting_quantile + quantile_adapt_rate * (torch
    .sigmoid(logits.mean()) - starting_quantile)

    # Calculate dynamic blending coefficient based on logits
    variability
    dynamic_blend_coeff = torch.sigmoid(logits_variability) *
    dynamic_blend_rate

    # Prepare components of the blended loss
    logistic_loss = -F.logsigmoid(self.beta * logits / temperature)
    exp_loss = torch.exp(-self.beta * logits * temperature)

    # Blend the losses dynamically
    losses = dynamic_blend_coeff * logistic_loss + (1 -
    dynamic_blend_coeff) * exp_loss
    return losses
```

$$f_{dbaql}(\beta\rho) = \sigma(\mathbb{V}\mathrm{ar}[\beta\rho/\tau]) \cdot f_{dpo}(\beta\rho/0.9) + (1 - \sigma(\mathbb{V}\mathrm{ar}[\beta\rho/\tau])) \cdot f_{exp}(\beta\rho \cdot 0.9) \tag{9}$$
$$\tau = 0.05 \tag{10}$$

## E.2 AQL: Adaptive Quantile Loss

MT-Bench Score: 7.953

```python
def adaptive_quantile_loss(
    self,
    policy_chosen_logps: torch.FloatTensor,
    policy_rejected_logps: torch.FloatTensor,
    reference_chosen_logps: torch.FloatTensor,
    reference_rejected_logps: torch.FloatTensor,
) -> torch.FloatTensor:
    percentile = 0.5  # Start with the median quantile
    moving_quantile_weight = 0.01  # Weight for updating the moving
    quantile
    pi_logratios = policy_chosen_logps - policy_rejected_logps
    ref_logratios = reference_chosen_logps - reference_rejected_logps
    logits = pi_logratios - ref_logratios

    moving_quantile = percentile + moving_quantile_weight * (torch.
    sigmoid(logits.mean()) - percentile)

    quantile_weights = torch.sigmoid(-self.beta * (logits -
    moving_quantile))

    logistic_losses = -F.logsigmoid(self.beta * logits)
    hinge_losses = torch.relu(1 - self.beta * logits)

    # Blend the logistic and hinge losses based on the dynamic
    quantile weight
    losses = quantile_weights * logistic_losses + (1 -
    quantile_weights) * hinge_losses
    return losses
```

$$f_{aql}(\beta\rho) = q \cdot f_{dpo}(\beta\rho) + (1 - q) \cdot f_{slic}(\beta\rho) \tag{11}$$

$$q = \sigma(\tau m_2 - \beta\rho) \tag{12}$$

$$m_2 = 0.5 + 0.01 \cdot \left(\mathbb{E}[\sigma(\beta\rho/\tau)] - 0.5\right) \tag{13}$$

$$\tau = 0.05 \tag{14}$$

## E.3 PADLL: Performance Adaptive Decay Logistic Loss

MT-Bench Score: 7.941

```python
def performance_adaptive_decay_logistic_loss(
    self,
    policy_chosen_logps: torch.FloatTensor,
    policy_rejected_logps: torch.FloatTensor,
    reference_chosen_logps: torch.FloatTensor,
    reference_rejected_logps: torch.FloatTensor,
) -> torch.FloatTensor:
    base_decay = 0.9
    mismatch_penalty = 0.5  # Penalty decay for mismatched choices

    pi_logratios = policy_chosen_logps - policy_rejected_logps
    ref_logratios = reference_chosen_logps - reference_rejected_logps
    logits = pi_logratios - ref_logratios
    mismatches = (logits < 0).float()  # Identify mismatches

    adaptive_decay = base_decay * (1 - mismatches * mismatch_penalty)
    weighted_losses = adaptive_decay * -F.logsigmoid(self.beta *
    logits)
    return weighted_losses
```

$$f_{padll}(\beta\rho) = \delta_{\text{adpt}} \cdot f_{dpo}(\beta\rho) \tag{15}$$

$$= \delta_{\text{base}} \cdot \left(1 - \mathbb{1}_{[\rho<0]} \cdot \tau\right) \cdot f_{dpo}(\beta\rho) \tag{16}$$

$$= \delta_{\text{base}} \cdot \left(1 - \mathbb{1}_{[\rho<0]} \cdot \tau\right) \cdot \log\left(1 + \exp(-\beta\rho)\right) \tag{17}$$

$$= 0.9 \cdot \left(1 - \mathbb{1}_{[\rho<0]} \cdot 0.5\right) \cdot \log\left(1 + \exp(-\beta\rho)\right) \tag{18}$$

This loss can also be rewritten as:

$$f_{padll}(\beta, \rho) = \begin{cases} \delta_{\text{pos}} \cdot f_{dpo}(\beta\rho), & \text{if } \rho \geq 0 \\ \delta_{\text{neg}} \cdot f_{dpo}(\beta\rho), & \text{if } \rho < 0 \end{cases} \quad , \text{where} \quad \delta_{\text{pos}} > \delta_{\text{neg}} > 0 \tag{19}$$

## E.4 AQFL: Adaptive Quantile Feedback Loss

MT-Bench Score: 7.931

```python
def adaptive_quantile_feedback_loss(
    self,
    policy_chosen_logps: torch.FloatTensor,
    policy_rejected_logps: torch.FloatTensor,
    reference_chosen_logps: torch.FloatTensor,
    reference_rejected_logps: torch.FloatTensor,
) -> torch.FloatTensor:
    import torch.nn.functional as F

    quantile_update_rate = 0.05
    distance_scale = 0.1

    pi_logratios = policy_chosen_logps - policy_rejected_logps
    ref_logratios = reference_chosen_logps - reference_rejected_logps
    logits = pi_logratios - ref_logratios
    logits_std = logits.std()

    adaptive_quantile = logits_std * torch.sigmoid(-logits).mean()
    adaptive_quantile += quantile_update_rate * (torch.sigmoid(logits.
    mean()) - adaptive_quantile)

    distance_from_quantile = (logits - adaptive_quantile).abs()
    blend_rate = torch.sigmoid(distance_scale * distance_from_quantile
    )

    logistic_losses = -F.logsigmoid(self.beta * logits)
    hinge_losses = torch.relu(1 - self.beta * logits)

    losses = blend_rate * logistic_losses + (1 - blend_rate) *
    hinge_losses
    return losses
```

$$f_{aqfl}(\beta\rho) = r \cdot f_{dpo}(\beta\rho) + (1 - r) \cdot f_{slic}(\beta\rho) \tag{20}$$

$$r = \sigma(0.1 * d) \tag{21}$$

$$d = |\beta\rho/\tau - m_2| \tag{22}$$

$$m_2 = m_1 + 0.05 \cdot \left(\sigma(\mathbb{E}[\beta\rho/\tau] - m_1)\right) \tag{23}$$

$$m_1 = \mathbb{E}[\sigma(-\beta\rho/\tau)] \cdot \sqrt{\text{Var}[\beta\rho/\tau]} \tag{24}$$

$$\tau = 0.05 \tag{25}$$

## E.5 CELL: Combined Exponential + Logistic Loss

MT-Bench Score: 7.925

```python
def combined_exp_logistic_loss(
    self,
    policy_chosen_logps: torch.FloatTensor,
    policy_rejected_logps: torch.FloatTensor,
    reference_chosen_logps: torch.FloatTensor,
    reference_rejected_logps: torch.FloatTensor,
) -> torch.FloatTensor:
    pi_logratios = policy_chosen_logps - policy_rejected_logps
    ref_logratios = reference_chosen_logps - reference_rejected_logps
    logits = pi_logratios - ref_logratios
    exp_losses = torch.exp(-self.beta * logits)
    log_losses = -F.logsigmoid(self.beta * logits)
    # Combine the losses with a tunable mixing coefficient
    alpha = 0.5
    losses = alpha * exp_losses + (1 - alpha) * log_losses
    return losses
```

$$f_{cell}(\beta\rho) = 0.5 \cdot f_{dpo}(\beta\rho) + 0.5 \cdot f_{exp}(\beta\rho) \tag{26}$$

### E.6    LRML: Log Ratio Modulated Loss

MT-Bench Score: 7.916

```python
def log_ratio_modulated_loss(
    self,
    policy_chosen_logps: torch.FloatTensor,
    policy_rejected_logps: torch.FloatTensor,
    reference_chosen_logps: torch.FloatTensor,
    reference_rejected_logps: torch.FloatTensor,
) -> torch.FloatTensor:
    pi_logratios = policy_chosen_logps - policy_rejected_logps
    ref_logratios = reference_chosen_logps - reference_rejected_logps
    logits = pi_logratios - ref_logratios
    # Modulate the mixing coefficient based on the log ratio
    magnitudes
    log_ratio_modulation = torch.sigmoid(logits)
    logistic_component = -F.logsigmoid(self.beta * logits)
    exp_component = torch.exp(-self.beta * logits)
    # Blend between logistic and exponential component based on log
    ratio modulation
    losses = logistic_component * (1 - log_ratio_modulation) +
    exp_component * log_ratio_modulation
    return losses
```

$$f_{lrml}(\beta\rho) = (1 - \sigma(\beta\rho/\tau)) \cdot f_{dpo}(\beta\rho) + \sigma(\beta\rho/\tau) \cdot f_{exp}(-\beta\rho) \tag{27}$$
$$\tau = 0.05 \tag{28}$$

### E.7    PFL: Policy Focused Loss

MT-Bench Score: 7.900

```python
def policy_focused_loss(
    self,
    policy_chosen_logps: torch.FloatTensor,
    policy_rejected_logps: torch.FloatTensor,
    reference_chosen_logps: torch.FloatTensor,
    reference_rejected_logps: torch.FloatTensor,
) -> torch.FloatTensor:
```

```
focus_scale = 2.0  # Scale to emphasize or de-emphasize based on
the correctness of predictions

pi_logratios = policy_chosen_logps - policy_rejected_logps
ref_logratios = reference_chosen_logps - reference_rejected_logps
logits = pi_logratios - ref_logratios
is_correct = policy_chosen_logps > policy_rejected_logps

logistic_losses = -F.logsigmoid(logits)
hinge_losses = torch.relu(1 - logits)

focused_loss = torch.where(
    is_correct,
    logistic_losses / focus_scale,  # De-emphasize correct
predictions
    hinge_losses * focus_scale  # Emphasize incorrect predictions
)
return focused_loss
```

Interestingly, the PFL generated function code did not include any $\beta$ values in the loss function. We have added it to the corrected code for the IMDb experiment, as well as to the mathematical expression below. We account for the consistency of the logit values by dividing with $\tau = 0.05$, which is the same value as $\beta$ in the discovery process.

$$f_{pfl}(\beta\rho) = 1/2 \cdot f_{dpo}(\beta\rho/\tau) \cdot \mathbb{1}_{[\pi_w > \pi_r]} + 2 \cdot f_{slic}(\beta\rho/\tau) \cdot \mathbb{1}_{[\pi_w \leq \pi_r]} \tag{29}$$
$$\tau = 0.05 \tag{30}$$

## F  Extended Related Work

In the following, we provide an extended related work of the related work included in the main paper.

**Evolution and Search with Large Language Models**. LLMs provide a fast and automated way to create multiple candidate solutions for a problem stated in natural language [Song et al., 2024], which makes them powerful tools for driving population-based search procedures. Various recent works have applied this approach to coding problems [Romera-Paredes et al., 2024], neural architecture search [Chen et al., 2024a, Holt et al., 2024b,d], virtual robotic design settings [Lehman et al., 2023], reward functions [Ma et al., 2023, Yu et al., 2023], and algorithm heuristics [Liu et al., 2024]. Finally, recently LLMs have shown to be capable of acting as recombination operators for black-box optimization with Evolution Strategies [Lange et al., 2024] and for Quality-Diversity approaches [Lim et al., 2024]. Additionally, prior work has shown LLM multi-agent frameworks' ability to excel at large code-generation tasks [Holt et al., 2024c], which we could envision enabling scaling up searching for larger candidate solutions for more complicated objectives or algorithms than the existing approaches that fit within the LLMs output context window.

**Automated Discovery for Machine Learning**. There are many other approaches to automating the discovery of generalizable machine learning algorithms. Some prior works explore the space of ML functions using genetic algorithms and a hand-crafted domain-specific language for reinforcement learning algorithms [Co-Reyes et al., 2021], curiosity algorithms [Alet et al., 2020], and optimizers [Chen et al., 2024b]. Other works instead parameterize a transferrable objective function using neural networks and optimize them with evolution strategies or meta-gradients. For example, Lu et al. [2022], Jackson et al. [2024b], Houthooft et al. [2018], Alfano et al. [2024], Kirsch et al. [2019], Oh et al. [2020], Jackson et al. [2024a] discover policy optimization objectives, Metz et al. [2022] evolves neural network optimizers, and Lange et al. [2023b,a] evolve blackbox optimizers. Moreover, automatically discovering closed-form functions (i.e., symbolic regression), works exist that leverage RL [Petersen et al., 2020], gradient descent [Kacprzyk et al., 2024], RL with evolution strategies [Mundhenk et al., 2021], pre-training transformers [Biggio et al., 2021] and hybrid combinations of pre-training transformers, which are further refined with RL and evolution strategies [Holt et al., 2023b]. Furthermore, more general-purpose ML approaches exist to perform sequential modeling of

processes [Holt et al., 2022], which can be readily used in model-based RL approaches [Holt et al., 2023a, 2024a].

**Preference Optimization Algorithms**. While the reduction to supervised learning makes DPO and alternatives easier to use, other approaches have sought to simplify the RL step, including using variants of REINFORCE [Ahmadian et al., 2024, Gemma-Team et al., 2024] as well as more fine-grained feedback [Wu et al., 2024] through preferences over individual steps in the reasoning process [Uesato et al., 2022, Lightman et al., 2023] or reward redistribution [Chan et al., 2024]. Others use iterative offline training interleaved with sampling from the policy model and obtaining a preference ranking from themselves [Xu et al., 2023], another judge LLM [Guo et al., 2024], or an oracle [Swamy et al., 2024].

# G   Full Run Log

We provide a full run below, formatted for readability.

```
==========
thought
Since the logistic log loss currently has the highest fitness and is a
    direct optimization of the log likelihood ratio, let's try a
   variation that includes a margin in the log likelihood ratios,
   inspired by the concept of a margin in SVM hinge loss. Introducing
    a margin could create a buffer that leads to more robust learning
   , as seen in SVMs.
name
logistic_margin_loss
code
def logistic_margin_loss(
    self,
    policy_chosen_logps: torch.FloatTensor,
    policy_rejected_logps: torch.FloatTensor,
    reference_chosen_logps: torch.FloatTensor,
    reference_rejected_logps: torch.FloatTensor,
) -> torch.FloatTensor:
    margin = 0.1
    pi_logratios = policy_chosen_logps - policy_rejected_logps
    ref_logratios = reference_chosen_logps - reference_rejected_logps
    logits_with_margin = pi_logratios - ref_logratios - margin
    losses = -F.logsigmoid(self.beta * logits_with_margin)
    return losses

==========
Fitness: 7.709375.
Please generate the next one.
==========
thought
The logistic margin loss outperformed the hinge and ipo losses but did
    not surpass the original logistic log loss. Perhaps introducing
   an adaptive margin based on the confidence in the predictions
   could yield improvements. The margin could be set proportional to
   the absolute difference between the policy's and the reference's
   log probabilities. This could potentially penalize incorrect
   confident predictions more heavily.
name
adaptive_margin_logistic_loss
code
def adaptive_margin_logistic_loss(
    self,
    policy_chosen_logps: torch.FloatTensor,
    policy_rejected_logps: torch.FloatTensor,
    reference_chosen_logps: torch.FloatTensor,
    reference_rejected_logps: torch.FloatTensor,
) -> torch.FloatTensor:
```

```
        confidence_margin = torch.abs(policy_chosen_logps -
        policy_rejected_logps) - torch.abs(reference_chosen_logps -
        reference_rejected_logps)
        pi_logratios = policy_chosen_logps - policy_rejected_logps
        ref_logratios = reference_chosen_logps - reference_rejected_logps
        logits_with_margin = pi_logratios - ref_logratios -
        confidence_margin
        losses = -F.logsigmoid(self.beta * logits_with_margin)
        return losses

==========
Fitness: 7.51875.
Please generate the next one.
==========
thought
The adaptive margin did not yield an improvement. It might have
    introduced too much variability or penalization. Let's go back to
    a fixed structure but combine elements from both logistic loss and
     hinge loss, with the idea of encouraging a larger margin when the
     decision is correct, while still having a soft, probabilistic
    interpretation. This combined loss could retain the benefits of
    both worlds.
name
combined_logistic_hinge_loss
code
def combined_logistic_hinge_loss(
    self,
    policy_chosen_logps: torch.FloatTensor,
    policy_rejected_logps: torch.FloatTensor,
    reference_chosen_logps: torch.FloatTensor,
    reference_rejected_logps: torch.FloatTensor,
) -> torch.FloatTensor:
    pi_logratios = policy_chosen_logps - policy_rejected_logps
    ref_logratios = reference_chosen_logps - reference_rejected_logps
    logits = pi_logratios - ref_logratios
    logistic_losses = -F.logsigmoid(self.beta * logits)
    hinge_losses = torch.relu(1 - self.beta * logits)
    combined_losses = logistic_losses + hinge_losses
    return combined_losses

==========
Fitness: 7.7375.
Please generate the next one.
==========
thought
The combined logistic and hinge loss slightly outperformed the
    baseline logistic log loss but still did not achieve a significant
     improvement. Considering the nature of RLHF, where we prefer the
    model to be correct but not overly confident, we can explore a
    scaled version of the combined loss where the logistic part is
    emphasized more heavily, as it intrinsically accounts for
    confidence due to its probabilistic nature.
name
scaled_combined_logistic_hinge_loss
code
def scaled_combined_logistic_hinge_loss(
    self,
    policy_chosen_logps: torch.FloatTensor,
    policy_rejected_logps: torch.FloatTensor,
    reference_chosen_logps: torch.FloatTensor,
    reference_rejected_logps: torch.FloatTensor,
) -> torch.FloatTensor:
    logistic_scale = 2.0  # Emphasize logistic loss more
    hinge_scale = 0.5    # De-emphasize hinge loss
```

```
    pi_logratios = policy_chosen_logps - policy_rejected_logps
    ref_logratios = reference_chosen_logps - reference_rejected_logps
    logits = pi_logratios - ref_logratios

    logistic_losses = logistic_scale * (-F.logsigmoid(self.beta *
    logits))
    hinge_losses = hinge_scale * (torch.relu(1 - self.beta * logits))

    combined_losses = logistic_losses + hinge_losses
    return combined_losses
```

==========
Fitness: 7.85625.
Please generate the next one.
==========
thought
The scaled combined loss has shown an improvement, which indicates
    that balancing the contribution between the logistic and hinge
    components is beneficial. To further expand on this idea, we could
     try a form of annealing where early in training we use more of
    the logistic component, encouraging better probability estimation,
     and later on a switch gradually to the hinge component,
    emphasizing decisions with a margin. We'll simulate this by using
    a weighting factor that changes linearly through the logits,
    giving lower logits (earlier, less confident predictions) a higher
     weight for the logistic loss.
name
annealed_combined_logistic_hinge_loss
code
```
def annealed_combined_logistic_hinge_loss(
    self,
    policy_chosen_logps: torch.FloatTensor,
    policy_rejected_logps: torch.FloatTensor,
    reference_chosen_logps: torch.FloatTensor,
    reference_rejected_logps: torch.FloatTensor,
) -> torch.FloatTensor:
    pi_logratios = policy_chosen_logps - policy_rejected_logps
    ref_logratios = reference_chosen_logps - reference_rejected_logps
    logits = pi_logratios - ref_logratios

    logistic_scale = torch.sigmoid(-logits)  # Higher weight for lower
     logits
    hinge_scale = torch.sigmoid(logits)      # Higher weight for
    higher logits

    logistic_losses = logistic_scale * (-F.logsigmoid(self.beta *
    logits))
    hinge_losses = hinge_scale * (torch.relu(1 - self.beta * logits))

    combined_losses = logistic_losses + hinge_losses
    return combined_losses
```

==========
Fitness: 7.66875.
Please generate the next one.
==========
thought
The annealed combined loss did not yield the expected improvement and
    performed worse than the scaled version. It's possible the
    transition from logistic to hinge was too aggressive. For the next
     iteration, let's return to a simpler concept and try a variation
    of the logistic loss with a squared term, similar to squared hinge
     loss, which might penalize incorrect predictions more smoothly
    than the linear term used in the standard logistic loss.
name

```
squared_logistic_loss
code
def squared_logistic_loss(
    self,
    policy_chosen_logps: torch.FloatTensor,
    policy_rejected_logps: torch.FloatTensor,
    reference_chosen_logps: torch.FloatTensor,
    reference_rejected_logps: torch.FloatTensor,
) -> torch.FloatTensor:
    pi_logratios = policy_chosen_logps - policy_rejected_logps
    ref_logratios = reference_chosen_logps - reference_rejected_logps
    logits = pi_logratios - ref_logratios
    logistic_losses = -F.logsigmoid(self.beta * logits)
    squared_losses = logistic_losses ** 2
    return squared_losses

==========
Fitness: 7.60062893081761.
Please generate the next one.
==========
thought
The squared logistic loss did not perform as well as hoped, possibly
    due to excessive penalization of difficult samples. This time, let
    's integrate a weighting mechanism that focuses on hard examples
    while using the logistic component. Inspired by the focal loss
    used in object detection, which gives more weight to harder,
    misclassified examples, we can apply a similar mechanism to
    emphasize learning from examples where the policy significantly
    differs from the reference.
name
focal_logistic_loss
code
def focal_logistic_loss(
    self,
    policy_chosen_logps: torch.FloatTensor,
    policy_rejected_logps: torch.FloatTensor,
    reference_chosen_logps: torch.FloatTensor,
    reference_rejected_logps: torch.FloatTensor,
) -> torch.FloatTensor:
    gamma = 2.0  # Focusing parameter for modulating the loss
    pi_logratios = policy_chosen_logps - policy_rejected_logps
    ref_logratios = reference_chosen_logps - reference_rejected_logps
    logits = pi_logratios - ref_logratios
    sigmoids = F.sigmoid(logits)
    focal_weights = (1 - sigmoids) ** gamma  # Focus more on harder
    examples
    logistic_losses = -focal_weights * F.logsigmoid(self.beta * logits
    )
    return logistic_losses

==========
Fitness: 7.840625.
Please generate the next one.
==========
thought
The focal logistic loss seems to have a moderate effect, indicating
    that prioritizing hard examples has some merit. To build on this,
    an alternative could be to apply temperature scaling to the logits
     before computing the logistic loss. Temperature scaling is often
    used in model calibration and can soften the probability
    distribution, which might work well with RLHF where overconfidence
     in predictions is undesirable.
name
temperature_scaled_logistic_loss
code
```

```
def temperature_scaled_logistic_loss(
    self,
    policy_chosen_logps: torch.FloatTensor,
    policy_rejected_logps: torch.FloatTensor,
    reference_chosen_logps: torch.FloatTensor,
    reference_rejected_logps: torch.FloatTensor,
) -> torch.FloatTensor:
    temperature = 2.0  # Temperature > 1.0 softens the logits
    pi_logratios = policy_chosen_logps - policy_rejected_logps
    ref_logratios = reference_chosen_logps - reference_rejected_logps
    tempered_logits = (pi_logratios - ref_logratios) / temperature
    logistic_losses = -F.logsigmoid(self.beta * tempered_logits)
    return logistic_losses
```

==========
Fitness: 7.86875.
Please generate the next one.
==========
thought
The temperature scaling seemed to have a positive impact, possibly due
    to better-calibrated probability estimates. To further explore
    this direction, we can try using label smoothing, which encourages
     the model not to be too confident about its predictions by
    preventing it from assigning full probability to a single class.
    It's a technique commonly used in classification tasks and might
    be beneficial for RLHF.
name
label_smoothed_logistic_loss
code
```
def label_smoothed_logistic_loss(
    self,
    policy_chosen_logps: torch.FloatTensor,
    policy_rejected_logps: torch.FloatTensor,
    reference_chosen_logps: torch.FloatTensor,
    reference_rejected_logps: torch.FloatTensor,
) -> torch.FloatTensor:
    label_smoothing = 0.1  # Epsilon for label smoothing
    pi_logratios = policy_chosen_logps - policy_rejected_logps
    ref_logratios = reference_chosen_logps - reference_rejected_logps
    logits = pi_logratios - ref_logratios
    smooth_positive = 1.0 - label_smoothing
    smooth_negative = label_smoothing / 2.0
    targets = torch.ones_like(logits) * smooth_positive
    losses = F.binary_cross_entropy_with_logits(
        self.beta * logits, targets, reduction='none'
    ) + smooth_negative * F.binary_cross_entropy_with_logits(
        -self.beta * logits, torch.zeros_like(logits), reduction='none
    ,
    )
    return losses.mean()
```

==========
Code not valid. Error:
Expected loss shape to be per input (e.g. (10,)), got torch.Size([])
Please generate the next one.
==========
thought
Since the label smoothing implementation had an issue with the loss
    shape being incorrect, it seems the loss reduction was erroneously
     computed over all inputs instead of keeping the per-input format.
     Let's adjust the implementation to make sure the loss retains the
     correct shape, by performing the mean operation separately for
    the positive and negative parts and then combining them, weighted
    appropriately.
name
```

```
corrected_label_smoothed_logistic_loss
code
def corrected_label_smoothed_logistic_loss(
    self,
    policy_chosen_logps: torch.FloatTensor,
    policy_rejected_logps: torch.FloatTensor,
    reference_chosen_logps: torch.FloatTensor,
    reference_rejected_logps: torch.FloatTensor,
) -> torch.FloatTensor:
    label_smoothing = 0.1  # Epsilon for label smoothing
    pi_logratios = policy_chosen_logps - policy_rejected_logps
    ref_logratios = reference_chosen_logps - reference_rejected_logps
    logits = pi_logratios - ref_logratios
    smooth_positive = 1.0 - label_smoothing
    smooth_negative = label_smoothing / 2.0
    positive_targets = torch.ones_like(logits) * smooth_positive
    negative_targets = torch.zeros_like(logits) * smooth_negative
    positive_losses = F.binary_cross_entropy_with_logits(
        self.beta * logits, positive_targets, reduction='none'
    )
    negative_losses = F.binary_cross_entropy_with_logits(
        self.beta * logits, negative_targets, reduction='none'
    )
    return (positive_losses + negative_losses) / 2

==========
Fitness: 6.425.
Please generate the next one.
==========
thought
The corrected label smoothing implementation didn't work as intended
    and significantly decreased the fitness. It seems that label
    smoothing may not align well with the objective of RLHF. In light
    of this, let's explore a different direction by introducing a
    decaying weight on older samples. The idea is to give higher
    importance to the more recent decisions made by the policy, under
    the assumption that they may be more aligned with the current
    state of the policy.
name
decaying_weights_logistic_loss
code
def decaying_weights_logistic_loss(
    self,
    policy_chosen_logps: torch.FloatTensor,
    policy_rejected_logps: torch.FloatTensor,
    reference_chosen_logps: torch.FloatTensor,
    reference_rejected_logps: torch.FloatTensor,
) -> torch.FloatTensor:
    decay_rate = 0.9  # Weight decay for older samples
    batch_size = policy_chosen_logps.size(0)
    decay_weights = decay_rate ** torch.arange(batch_size - 1, -1, -1)
    decay_weights = decay_weights.to(policy_chosen_logps.device)

    pi_logratios = policy_chosen_logps - policy_rejected_logps
    ref_logratios = reference_chosen_logps - reference_rejected_logps
    logits = pi_logratios - ref_logratios
    losses = decay_weights * -F.logsigmoid(self.beta * logits)
    return losses / decay_weights.sum()  # Normalizing by sum of
    weights

==========
Fitness: 7.871875.
Please generate the next one.
==========
thought
```

While the decaying weights logistic loss provided a slight increase in
    fitness, it suggests that emphasizing more recent samples can be
    beneficial, but the approach might need some refinement. We could
    potentially improve this by making the decay adaptive based on the
    performance of each choice. The idea would be to give less weight
    to choices that are heavily mismatched with the reference, under
    the hypothesis that these could be outliers or errors.
name
performance_adaptive_decay_logistic_loss
code

```
def performance_adaptive_decay_logistic_loss(
    self,
    policy_chosen_logps: torch.FloatTensor,
    policy_rejected_logps: torch.FloatTensor,
    reference_chosen_logps: torch.FloatTensor,
    reference_rejected_logps: torch.FloatTensor,
) -> torch.FloatTensor:
    base_decay = 0.9
    mismatch_penalty = 0.5  # Penalty decay for mismatched choices

    pi_logratios = policy_chosen_logps - policy_rejected_logps
    ref_logratios = reference_chosen_logps - reference_rejected_logps
    logits = pi_logratios - ref_logratios
    mismatches = (logits < 0).float()  # Identify mismatches

    adaptive_decay = base_decay * (1 - mismatches * mismatch_penalty)
    weighted_losses = adaptive_decay * -F.logsigmoid(self.beta *
    logits)
    return weighted_losses
```

==========
Fitness: 7.940625.
Please generate the next one.
==========
thought
The performance-adaptive decay approach provided a slight improvement
    in fitness, suggesting that dynamically adjusting the loss based
    on the correctness of predictions is a promising direction. To
    take this further, let's create a hybrid approach which combines
    the performance-adaptive decay with a form of margin-based loss.
    This will aim to reduce the weights of not only the incorrect
    predictions but also those that are correct yet lack confidence,
    thereby promoting a more decisive policy.
name
hybrid_performance_margin_decay_logistic_loss
code

```
def hybrid_performance_margin_decay_logistic_loss(
    self,
    policy_chosen_logps: torch.FloatTensor,
    policy_rejected_logps: torch.FloatTensor,
    reference_chosen_logps: torch.FloatTensor,
    reference_rejected_logps: torch.FloatTensor,
) -> torch.FloatTensor:
    base_decay = 0.9
    margin = 0.2
    mismatch_penalty = 0.5  # Penalty decay for mismatched choices

    pi_logratios = policy_chosen_logps - policy_rejected_logps
    ref_logratios = reference_chosen_logps - reference_rejected_logps
    logits = pi_logratios - ref_logratios
    margin_logits = logits - margin
    mismatches = (margin_logits < 0).float()  # Identify mismatches
    with margin

    adaptive_decay = base_decay * (1 - mismatches * mismatch_penalty)
```

```
    weighted_losses = adaptive_decay * -F.logsigmoid(self.beta *
    margin_logits)
    return weighted_losses

==========
Fitness: 7.6125.
Please generate the next one.
==========
thought
The hybrid approach with the performance margin decay did not achieve
    the desired effectiveness. Introducing a margin may have been too
    punitive on correct predictions that are close to the boundary. To
     maintain the balance, let's explore the idea of using a triplet-
    style loss, which is popular in embedding learning. By treating
    the chosen policy logs as the anchor and the reference chosen and
    rejected logs as positive and negative examples, respectively, we
    can encourage the chosen policy decisions to be closer to the
    reference chosen decisions and farther away from the reference
    rejected decisions.
name
triplet_style_logistic_loss
code
def triplet_style_logistic_loss(
    self,
    policy_chosen_logps: torch.FloatTensor,
    policy_rejected_logps: torch.FloatTensor,
    reference_chosen_logps: torch.FloatTensor,
    reference_rejected_logps: torch.FloatTensor,
) -> torch.FloatTensor:
    alpha = 0.1  # Margin for the triplet loss
    positive_distance = -F.logsigmoid(self.beta * (policy_chosen_logps
     - reference_chosen_logps))
    negative_distance = -F.logsigmoid(self.beta * (policy_chosen_logps
     - reference_rejected_logps))
    triplet_loss = F.relu(positive_distance - negative_distance +
    alpha)
    return triplet_loss

==========
Code not valid. Error:
isnan(): argument 'input' (position 1) must be Tensor, not NoneType
Please generate the next one.
==========
thought
It seems there was an error in the triplet-style loss function, which
    suggests that the computation may have been incorrect or
    incomplete. To correct this, we need to ensure that all
    intermediate steps yield non-empty tensors. Triplet loss can still
     be a useful approach as it inherently contains the concept of a
    margin while allowing for a dynamic balance between the chosen and
     rejected log probabilities. Let's revise the code with added
    checks to prevent any NoneType issues and ensure that the loss is
    computed correctly.
name
revised_triplet_style_logistic_loss
code
def revised_triplet_style_logistic_loss(
    self,
    policy_chosen_logps: torch.FloatTensor,
    policy_rejected_logps: torch.FloatTensor,
    reference_chosen_logps: torch.FloatTensor,
    reference_rejected_logps: torch.FloatTensor,
) -> torch.FloatTensor:
    alpha = 0.1  # Margin for the triplet loss
    # Compute distances
```

```
        positive_distance = policy_chosen_logps - reference_chosen_logps
        negative_distance = policy_chosen_logps - reference_rejected_logps
        # Calculate the triplet loss
        triplet_loss = F.relu(positive_distance - negative_distance +
        alpha)
        # Convert triplet loss into a logistic-style loss
        logistic_triplet_loss = -F.logsigmoid(self.beta * triplet_loss)

        return logistic_triplet_loss.mean()  # Ensure the loss is averaged
```

==========
Code not valid. Error:
Expected loss shape to be per input (e.g. (10,)), got torch.Size([])
Please generate the next one.
==========
thought
The revised triplet-style logistic loss still encountered an issue,
    likely due to improper application of the operations and reduction
     at the end, which should be on a per-example basis. Triplet loss
    typically works with distances embedded in a space, and here we
    are dealing with log probabilities, so the direct application may
    not be correct. For the next attempt, let's simplify: we'll
    reinforce the relationship between the chosen and rejected log
    probabilities directly by using their difference, promoting a
    wider margin while still within the logistic loss framework.
name
reinforced_margin_logistic_loss
code

```
def reinforced_margin_logistic_loss(
    self,
    policy_chosen_logps: torch.FloatTensor,
    policy_rejected_logps: torch.FloatTensor,
    reference_chosen_logps: torch.FloatTensor,
    reference_rejected_logps: torch.FloatTensor,
) -> torch.FloatTensor:
    margin = 0.1  # Margin introduced for reinforcing the difference
    # Calculate log probability differences
    chosen_difference = policy_chosen_logps - reference_chosen_logps
    rejected_difference = policy_rejected_logps -
    reference_rejected_logps
    # Apply the margin to differences
    reinforced_chosen = F.relu(chosen_difference + margin)
    reinforced_rejected = F.relu(rejected_difference - margin)
    # Compute logistic losses with reinforced margins
    losses = -F.logsigmoid(self.beta * reinforced_chosen) - F.
    logsigmoid(-self.beta * reinforced_rejected)
    return losses.mean(0)  # Maintain loss shape as per input
```

==========
Code not valid. Error:
Expected loss shape to be per input (e.g. (10,)), got torch.Size([])
Please generate the next one.
==========
thought
There was an error in the implementation of the reinforced margin
    logistic loss, likely because the mean operation was again placed
    incorrectly. The loss should be returned without any aggregation
    to maintain the per-input structure. Keeping this in mind, we need
     to correctly apply a margin to reinforce the separation between
    policy decisions and reference decisions.
name
corrected_reinforced_margin_logistic_loss
code

```
def corrected_reinforced_margin_logistic_loss(
    self,
```

```
        policy_chosen_logps: torch.FloatTensor,
        policy_rejected_logps: torch.FloatTensor,
        reference_chosen_logps: torch.FloatTensor,
        reference_rejected_logps: torch.FloatTensor,
) -> torch.FloatTensor:
        margin = 0.1  # Margin for reinforcing the separation
        # Calculate log probability differences
        chosen_difference = policy_chosen_logps - reference_chosen_logps
        rejected_difference = policy_rejected_logps -
        reference_rejected_logps
        # Reinforce chosen logits by adding a margin
        reinforced_chosen = chosen_difference + margin
        # Reinforce rejected logits by subtracting a margin
        reinforced_rejected = rejected_difference - margin
        # Compute logistic losses for reinforced logits
        chosen_losses = -F.logsigmoid(self.beta * reinforced_chosen)
        rejected_losses = -F.logsigmoid(-self.beta * reinforced_rejected)
        # Combine losses without applying mean
        return chosen_losses + rejected_losses

==========
Fitness: 7.525.
Please generate the next one.
==========
thought
The approach of applying a static margin did not yield the expected
    improvements. It's worth considering a different perspective;
    instead of focusing on introducing margins or applying decays, let
    's attempt to directly optimize the policy's certainty.
    Intuitively, the policy should be more confident when choosing
    actions similar to the reference and less confident otherwise. A
    potential approach would be to scale the logistic loss by the
    absolute difference between the policy's choice and the reference
    choice, thus directly tying the loss to the policy's certainty in
    its decision.
name
certainty_scaled_logistic_loss
code
def certainty_scaled_logistic_loss(
        self,
        policy_chosen_logps: torch.FloatTensor,
        policy_rejected_logps: torch.FloatTensor,
        reference_chosen_logps: torch.FloatTensor,
        reference_rejected_logps: torch.FloatTensor,
) -> torch.FloatTensor:
        # Compute the absolute certainty differences
        chosen_certainty_diff = torch.abs(policy_chosen_logps -
        reference_chosen_logps)
        rejected_certainty_diff = torch.abs(policy_rejected_logps -
        reference_rejected_logps)
        # Calculate the mean certainty difference
        mean_certainty_diff = (chosen_certainty_diff +
        rejected_certainty_diff) / 2
        # Compute the logistic loss
        losses = -F.logsigmoid(self.beta * (policy_chosen_logps -
        policy_rejected_logps))
        # Scale the loss by the certainty difference
        certainty_scaled_losses = losses * mean_certainty_diff
        return certainty_scaled_losses

==========
Fitness: 7.33125.
Please generate the next one.
```

