# OpenReview forum: "Discovering Preference Optimization Algorithms with and for Large Language Models"
_NeurIPS.cc/2024/Conference — NeurIPS 2024 poster_

### Official Review · Reviewer_gRRA · 2024-07-13

**Soundness:** 3
**Presentation:** 3
**Contribution:** 3
**Rating:** 6
**Confidence:** 3

**Summary:**

This paper introduces a method of searching for offline RL objectives by using LLMs to generate and refine objectives. They demonstrate that several objectives discovered using this method are able to achieve higher evaluation scores than existing objectives (e.g. DPO) on a variety of benchmark tests. They provide a high level analysis of one discovered objective, hypothesizing that it may possess desirable properties that lead to these performance boosts.

**Strengths:**

The paper is well written, organized, and easy to understand. While the use of LLMs to modify prompts and self-improve in other ways has been studied, using them to improve objective functions is a novel idea that the authors demonstrate holds potential. The authors demonstrate that the objectives discovered can be useful, showing improvements on a variety of generation and classification tasks.

**Weaknesses:**

The main weakness of this paper in my opinion is that it isn't clear what results the paper is trying to present. While this is partially an issue of organization, it is also an issue with the amount of evidence given to the claims in the paper. If these was made clearer, I would be inclined to raise my score. I detail below:

1. The paper starts by introducing a new method for discovering new optimization objectives using LLMs, then transitions quite abruptly to describing the properties of DiscoPop. While both could be valuable contributions, I don't feel that either is given quite the attention required. It would help to clarify what the main focus of the paper is. If DiscoPOP is the focus, then claims about its properties should be better studied. If the discovery method is the focus, more emphasis should be placed on the method's performance and ability to produce novel and useful objectives.

2. The majority of insights about DiscoPOP are presented as hypotheses or intuitive insights (e.g. the non-convex portion, and how the model behaves at $\pm\inf$. This is not enough to support the claim made in the conclusion that they provide insights into what properties an optimal objective function should possess. While it \emph{may} be enough to say the provide intuition into why DiscoPOP is successful, further experimentation would be necessary to make optimality claims like this.

3. Some portions of the discovery method are vague as well. Though the authors do not observe the models simply regurgitating old objectives and cite a pattern of objective refinement for supervised learning, this is of course not guaranteed, and not well studied. Further experiments or details on the how often they observe refinement vs regurgitation would help to support the utility of the method.

**Questions:**

1. How often are unit tests for the generated code failed?

**Limitations:**

The limitations described cover most points, including the difficulties and uncertainties associated with using LLMs to improve upon objectives in a rigorous fashion.

---

> ### Author Rebuttal · Authors · 2024-08-05
>
> We thank the reviewer for their thoughtful and extremely helpful review. We’re glad that the reviewer finds our approach to be both novel and useful, though we agree that more analysis would improve the paper significantly.
>
> > The majority of insights about DiscoPOP are presented as hypotheses or intuitive insights...further experimentation would be necessary
>
> This is a great point and we’re grateful to the reviewer for bringing it up. We’ve performed further analysis that *considerably* improves the paper and our understanding of the loss function. Previously, we merely hypothesized that the local optimum of the DiscoPOP loss could catch noisy or incorrect data points. We now have empirical evidence for this and include the results in the rebuttal PDF.
>
> In short, we identified which data points end up between the local optima after training, and found that 1.35% of points fall there (see Figure 1 in the PDF, where they are clearly visible). Although we use the binary preference labels from the IMDb dataset for training, the dataset also includes a reward score for each completion. When we analyze the data points that are between the local optima, we find that the positive and negative completions are *significantly* closer in absolute reward difference than the data points outside the local optimum (See Table 1 in PDF). This implies that the preference labels on those points are more difficult to distinguish and helps empirically validate our hypothesis. Thanks to the reviewers, we will be adding this analysis to the paper.
>
> >Further experiments or details on...refinement vs regurgitation
>
> This was another great suggestion. We’ve included an ablation on the CIFAR-10 results in which we *don’t* provide fitness feedback. Thus, the model is unable to perform refinement (as it does not know what performs well) and can only regurgitate ideas. This is a key baseline to compare to to validate our discovery method.
>
> In the attached rebuttal PDF, you can see that, without the fitness score, the LLM is unable to refine its ideas and can only regurgitate, thus leading to fewer steps of improvement.
>
> > On LLM Discovery vs. DiscoPOP as the main focus
>
> Our main focus was on DiscoPOP, as that is where the majority of our resources and efforts were focused. We will adjust the writing to make this clearer.
>
> We believe both parts are significant contributions of our paper, though we understand both could use more analysis. Thanks to the reviewer’s feedback, we’ve managed to significantly strengthen evidence for both.
>
> > what properties an optimal objective function should possess
>
> Thanks for bringing this up. This was imprecise language on our end.  We agree, and have now removed any claims about what an "optimal" objective should possess, instead replacing the word optimal with the word "good" in the conclusion (line 277 and in line 110).
>
> > How often are unit tests for the generated code failed?
>
> For GPT-4 the unit tests failed ~5% of the time; however, the model usually fixes it upon the feedback. We briefly experimented with weaker models and found the failure rates to be significantly higher.
>
> ---
> *We hope that most of the reviewer’s concerns have been addressed and, if so, they would consider updating their score. We’d be happy to engage in further discussions.*

---

> > ### Comment · Reviewer_gRRA · 2024-08-12
> >
> > Apologies for my late reply and thank you for your thorough response and clarifications. After going over your main response and supplemental PDF, I'm going to raise my score. The additional experiments on DiscoPOP properties and the experiment regarding refinement have satisfied my biggest concerns. The CIFAR-10 experiment is a good demonstrative example, though it doesn't prove that this will always be the case for the method, I believe it is a sufficiently convincing example to improve the point you were making in that section.

---

> ### Author Response · Authors · 2024-08-10
>
> We would like to once again thank the reviewer for their time and feedback. We've incorporated it into our manuscript and we believe it has strengthened our paper.
>
> We hope our rebuttal, which includes substantial additional results and analysis, has addressed the reviewer's concerns.
>
> Seeing as the discussion period is coming to a close, could the reviewer please let us know if they have any further questions or concerns about our submission?

---

### Official Review · Reviewer_663E · 2024-07-13

**Soundness:** 3
**Presentation:** 3
**Contribution:** 3
**Rating:** 6
**Confidence:** 3

**Summary:**

The paper proposes DiscoPOP, an algorithm for discovering preference optimization loss functions using Large Language Models (LLMs). The authors propose an LLM-driven objective discovery process by iterative prompting LLMs by previously evaluated performance metrics. Experiments on various benchmarks demonstrate its effectiveness.

**Strengths:**

1. This method is innovative. The idea of updating the loss function through automated exploration is both interesting and novel. The process being fully automated and end-to-end makes it particularly neat.
2. The results are impressive. DiscoPOP demonstrates strong performance across various benchmarks, showing its potential.

**Weaknesses:**

I think the updating process might be sensitive to the prompts used for proposing new loss functions.

**Questions:**

Refer to the weakness

**Limitations:**

Refer to the weakness

---

> ### Author Rebuttal · Authors · 2024-08-05
>
> We would like to thank the reviewer for their concise feedback. We’re happy that the reviewer finds the paper innovative and the results impressive. We understand the reviewer has concerns around prompt sensitivity. We would like to point the reviewer to a feel ablations we’ve run on this.
>
> In Appendix D3 of the paper, we show that the discovery process is very robust to different sampling parameters, prompts, and techniques from the LLM. We find little change in performance or behavior when we change the sampling temperature (unless it is set unreasonably high), telling it to “think” first, or sorting the input context.
>
> Notably, in our attached rebuttal PDF we show that providing the fitness in context is important to allow the LLM to refine its proposed ideas.
>
> ---
> *We hope that most of the reviewer’s concerns have been addressed and, if so, they would consider updating their score. We’d be happy to engage in further discussions.*

---

> > ### Comment · Reviewer_663E · 2024-08-13
> >
> > Hi,
> > I've read the rebuttal and I will keep my score as positive. Thanks.

---

### Official Review · Reviewer_MbcT · 2024-07-15

**Soundness:** 3
**Presentation:** 3
**Contribution:** 3
**Rating:** 7
**Confidence:** 3

**Summary:**

The paper proposes an algorithm to discover preference objective functions using LLM for LLM preference optimization. Authors conduct experiments with the discovered objectives on multiple datasets and demonstrate that the discovered objective functions generally perform better than baselines. The authors also show interesting insight into the discovered objective function, LRML.

**Strengths:**

1. The paper introduces an interesting and new LLM-driven objective discovery algorithm to search for good objective functions using LLMs for the preference optimization of LLMs.
2. The method successfully finds an objective function that can generally offer a better performance on multiple benchmarks, indicating the effectiveness of the discovery method.
3. The authors show interesting insights into the best objective function found by the proposed method.

**Weaknesses:**

1. The objective function is discovered with a different placement of $\beta$ than the objective function used for evaluation, which causes a misalignment of the discovery and the evaluation process. It would be nice to keep them aligned.
2. The LLM in the discovery process is a clever generator for searching candidates. It is unclear if the LLM's capability used in the discovery process can change the duration for finding a good objective function or if it will cause the search process to fail.
3. Similar to the point above, it would be nice to test some non-LLM-based generators of the objective function and perform some traditional searching algorithms such as evolutionary search to demonstrate the effectiveness of the LLM.

**Questions:**

See weakness above.

**Limitations:**

See weakness above.

---

> ### Author Rebuttal · Authors · 2024-08-05
>
> We thank the reviewer for their thorough and in-depth review. We are pleased that the reviewer finds our approach interesting and effective. We address the weaknesses outlined by the reviewer individually below.
>
> > The objective function is discovered with a different placement of $\beta$
>
> Thank you for highlighting this. We assume that the reviewer is referring to discrepancies between the code for the LLM sampler in Appendix E.6 and the formula presented in Equation (28), as explained in Appendix E. As a result of the reviewer’s comment, we found have now corrected a typo in the LRML formula, which now reads:
>
> $$ f_{lrml}(\beta\rho) = (1-\sigma(\beta\rho/\tau)) \cdot f_{dpo}(\beta\rho) + \sigma(\beta\rho/\tau) \cdot f_{exp}(-\beta\rho) $$
>
> The error occurred in the written mathematical notation, and the implementation remains the same.
>
> We introduce the $\tau$ parameter largely to retain the original semantic meaning of “beta” used in most papers, which is to adjust the KL-divergence regularisation. Since we use $\tau = 0.05$ they are equivalent.
>
> We have now corrected the mathematical notation for DBAQL, AQL, AQFL, and PFL as well.
>
> >  It is unclear if the LLM's capability used in the discovery process can change the duration for finding a good objective function or if it will cause the search process to fail.
>
> This is a good point. We briefly tried using LLama3 8b-instruct and found that it rarely responded with the correct format and working code, rendering it largely unusable. Should the reviewer request, we can do a more thorough sweep over base models. However, we do not expect this to fundamentally change our results.
>
> > it would be nice to...perform some traditional searching algorithms
>
> Unfortunately, this exact experiment would be very hard to implement, as we would need to design a domain-specific language to search over for the objective function or parameterize it in some way. Parameterizing the objective with a neural network is possible, but would require far too many inner loop training runs to be optimized. However, inspired by the reviewer’s feedback, we’ve designed an experiment that is similar in spirit to validate the effectiveness of the LLM-Driven Discovery method. In particular, we’ve added a version of the CIFAR-10 experiment where we do not return the fitness to the LLM. We show the results in Figure 2 of the attached PDF and confirm that, without the fitness, the LLM is unable to refine its ideas, as it does not have knowledge of which ones worked.
>
> ---
> *We hope that most of the reviewer’s concerns have been addressed and, if so, they would consider updating their score. We’d be happy to engage in further discussions.*

---

### Official Review · Reviewer_ePhP · 2024-07-15

**Soundness:** 3
**Presentation:** 3
**Contribution:** 3
**Rating:** 7
**Confidence:** 4

**Summary:**

This paper proposes a novel approach to improving LLMs by using an automated system to discover new optimization algorithms. Traditionally, enhancing LLMs has relied heavily on hand-designed loss functions, but this research employs an LLM to iteratively generate and refine these functions itself. This paper introduced DiscoPOP, which blends logistic and exponential losses to outshine existing methods. This algorithm was evaluated across a range of tasks, including multiturn dialogue, sentiment generation, and summarization, where it consistently delivered good results.

**Strengths:**

- The automatic exploration and evaluation pipeline is promising and it can discover new algorithms without human intervention.
- By using the above pipeline, the new discovered algorithm in preference optimization achieved SOTA, proving its effectiveness.
- This can contribute to lots of other area and let machines themselves to discover and evaluate new efficent algorithms.

**Weaknesses:**

Although this paper mianly focuses on preference optimization task, the dicovery method seems to be easily to adapt to other domains/tasks. A general concern with the paper is its reliance on a complex method that involves detailed tuning of hyperparameters. This complexity could limit the method's broader applicability unless the process can be generalized or adapted effectively to other contexts. However, the paper does not thoroughly address how to establish and optimize the discovery pipeline across different scenarios, which could hinder its practical utility. More guidance on adapting the methodology to a variety of use cases would significantly enhance its value and impact.

**Questions:**

1. How well does the proposed LLM-driven discovery method generalize to other domains/tasks beyond preference optimization? Are there specific modifications needed to adapt this method to other fields?
2. The authors mention that DiscoPOP blends logistic and exponential losses. Could you elaborate on the theoretical justification for this choice? How does this combination affect the convergence properties of the algorithm?
3. Considering the non-convex nature of DiscoPOP, what strategies do you suggest to avoid local minima during optimization?

**Limitations:**

The authors discussed the limitations of DiscoPOP.

---

> ### Author Rebuttal · Authors · 2024-08-05
>
> We thank the reviewer for their constructive feedback. We’re glad the reviewer finds the approach and results promising and effective.
>
> > does proposed LLM-driven discovery method generalize to other domains?
>
> In our submission, we demonstrated that it works well on CIFAR-10 in the small case study in Figure 2, where it discovers loss functions that outperform standard cross-entropy loss and transfers to other models and hyperparameters. We will adjust the writing to make this clearer.
>
> There were no significant modifications needed to adapt this method to other fields, the only change needed is to adapt the text in the prompt (Appendix A) to contain information and examples for the new setting.
> We chose preference optimization, as it is a new exciting area with great potential for novelty, though we agree that adapting our LLM-driven discovery method to other domains/tasks is an exciting future direction of work.
>
> > reliance on a complex method that involves detailed tuning...could limit the method's applicability unless the process can be generalized
>
> We’re not sure exactly which hyperparameters the reviewer is referring to, but Algorithm 1 does not contain any hyperparmaeters and is very easy to apply to other domains.
>
> >Theoretical justifications for DiscoPOP
>
> To understand DiscoPOP more from a theoretical perspective, we can closely compare it to the standard DPO loss. At the limits (eg when the relative preferences $\rho$ are very high or low), it has similar behavior to DPO. However, it mostly differs by its non-convex region, which we analyze in the rebuttal PDF. Previously, we merely hypothesized that the local optimum of the DiscoPOP loss could catch noisy or incorrect data points. We now have empirical evidence for this.
>
> In short, we identified which data points end up between the local optima after training, and found that 1.35% of points fall there (see Figure 1 in the PDF, where they are clearly visible). Although we use the binary preference labels from the IMDb dataset for training, the dataset also includes a reward score for each completion. When we analyze the data points that are between the local optima, we find that the positive and negative completions are *significantly* closer in absolute reward difference than the data points outside the local optimum (See Table 1 in PDF). This implies that the preference labels on those points are more difficult to distinguish and helps empirically validate our hypothesis. Thanks to the reviewers, we will be adding this analysis to the paper.
>
> We would like to further emphasize that, in the current offline preference optimization paradigm, training and evaluation differ significantly. In most settings in machine learning, the loss usually corresponds directly with some desired metric, such as the accuracy. However, in our setting,  while we train on an offline set of preference pairs, we ultimately evaluate the model using MT-Bench and GPT-4 as a judge. Thus, it’s not very clear how the loss function used necessarily corresponds to the downstream task. In fact, recent works [1, 2] show that optimizing the DPO loss function too much can lower the quality of the model. Thus, “theoretical” justifications are not as useful in this setting.
>
> > what strategies do you suggest to avoid local minima during optimization?
>
> We only found the local minima to be a problem when we set $\beta$ to be very low ($\leq 0.01$). Note that, while local minima are a problem when the loss function directly corresponds to the task, it’s not immediately clear if that’s the case in offline preference optimization (see our comment on this above).
>
> [1] Feng, Duanyu, et al. "Towards analyzing and understanding the limitations of dpo: A theoretical perspective." arXiv preprint arXiv:2404.04626 (2024).
>
> [2] Chen, Angelica, et al. "Preference Learning Algorithms Do Not Learn Preference Rankings." arXiv preprint arXiv:2405.19534 (2024).
>
> ---
> *We hope that most of the reviewer’s concerns have been addressed and, if so, they would consider updating their score. We’d be happy to engage in further discussions.*

---

> > ### Comment · Reviewer_ePhP · 2024-08-12
> >
> > Thanks for the reply. As you said, "*the only change needed is to adapt the text in the prompt*", I'm curious how to find the best prompt for a specific task. Are there any rules to follow?

---

> > > ### Author Response · Authors · 2024-08-12
> > >
> > > We didn't try a large number of prompts. We recommend just describing the setting in text (eg what the inputs and outputs of the function are) and adding a few examples of the code it should generate (along with their associated performance).

---

> > > > ### Comment · Reviewer_ePhP · 2024-08-12
> > > >
> > > > Thanks for the clarification. I don't have further questions. I will rise the score to 7.

---

> > > > > ### Author Response · Authors · 2024-08-12
> > > > >
> > > > > Thanks! Let us know if you have any other questions or concerns.

---

### Author Rebuttal · Authors · 2024-08-05

We are grateful to the reviewers for their insightful feedback. There is broad consensus amongst the reviewers that our approach is novel and effective.


$\color{red} R1$ (ePhP): “the new discovered algorithm in preference optimization achieved SOTA, proving its effectiveness.”

$\color{green} R2$ (MbcT): “The method successfully finds an objective function that...offer a better performance...indicating the effectiveness of the discovery method.”

$\color{blue} R3$ (663E): “This method is innovative...The results are impressive”

$\color{magenta} R4$ (gRRA): “using [LLM’s] to improve objective functions is a novel idea that the authors demonstrate holds potential”

Reviewers understandably had concerns about the lack of in-depth analysis of the DiscoPOP loss ($\color{red} R1$, $\color{magenta} R4$) function and its theoretical implications ($\color{green} R2$, $\color{magenta} R4$).

Reviewers also had concerns about the effectiveness ($\color{green} R2$, $\color{magenta} R4$) of LLM-driven discovery and its sensitivity ($\color{blue} R3$) to the prompt.

We address these concerns below.

## Deeper Analysis of DiscoPOP

We would like to deeply thank the reviewers for bringing this up. We’ve performed further analysis that *considerably* improves the paper and our understanding of the loss function. Previously, we merely hypothesized that the local optimum of the DiscoPOP loss could catch noisy or incorrect data points. We now have some empirical evidence for this.

In short, we identified which data points end up between the local optima after training, and found that 1.35% of points fall there (see Figure 1 in the PDF, where they are clearly visible). Although we use the binary preference labels from the IMDb dataset for training, the dataset also includes a reward score for each completion. When we analyze the data points that are between the local optima, we find that the positive and negative completions are *significantly* closer in absolute reward difference than the data points outside the local optimum (See Table 1 in PDF). This implies that the preference labels on those points are more difficult to distinguish and helps empirically validate our hypothesis. Thanks to the reviewers, we will be adding this analysis to the paper.

### Misc:

Thanks to a comment from $\color{green} R2$, we discovered a typo in equations (4) and (5) for DiscoPOP. We have since fixed this.

## More Analysis of LLM-Driven Discovery

We’ve added a key baseline experiment to our results to validate the effectiveness of the LLM-Driven Discovery method. In particular, we’ve added a version of the CIFAR-10 experiment where we do not return the fitness to the LLM. We show the results in Figure 2 of the attached PDF and confirm that, without the fitness, the LLM is unable to refine its ideas, as it does not have knowledge of which ones worked.

---

### Decision · Program_Chairs · 2024-09-25

**Decision:**

Accept (poster)

**Comment:**

The paper proposes a novel method that uses LLM to discover preference optimization algorithms. This falls into the metalearning category. The reviewers found the performance to be effective and the experiments to be comprehensive. During the rebuttal, the authors successfully addressed the reviewers' questions and concerns such as hyperparameters and the comprehensiveness of the experiments. I think it would be nice to provide the readers a more comprehensive related work section for the metalearning papers that were published before 2020.